**Measurement report: Distinct Emissions and Volatility Distribution of Intermediate Volatility Organic Compounds from on-road Chinese Gasoline Vehicle: Implication of High Secondary Organic Aerosol Formation Potential**

Rongzhi Tang[1,2,3,#], Quanyang Lu[4,#], Song Guo[1,3,*], Hui Wang[1], Kai Song[1], Ying Yu[1], Rui Tan[1], Kefan Liu[1], Ruizhe Shen[1], Shiyi Chen[1], Limin Zeng[1], Spiro D. Jorga[4], Zhou Zhang[5], Wenbin Zhang[5], Shijin Shuai[5], Allen L. Robinson[4,*]

*1 State Key Joint Laboratory of Environmental Simulation and Pollution Control, International Joint Laboratory for Regional Pollution Control, Ministry of Education (IJRC), College of Environmental Sciences and Engineering, Peking University, Beijing 100871, China P. R.*

*2 School of Environmental and Material Engineering, Yantai University, Yantai, 264003, PR China*

*3 Collaborative Innovation Center of Atmospheric Environment and Equipment Technology, Nanjing University of Information Science & Technology, Nanjing 210044, China P. R.*

*4 Department of Mechanical Engineering and Center for Atmospheric Particle Studies, Carnegie Mellon University, 5000 Forbes Avenue, Pittsburgh, Pennsylvania 15213, United States*

*5 State Key Laboratory of Automotive Safety and Energy, School of Vehicle and Mobility, Tsinghua University, Beijing, 100084, PR China*

Correspondence to: S. Guo songguo@pku.edu.cn, A. Robinson:alr@andrew.cmu.edu

[#]These authors contribute equally to this work

**Abstract**

In the present work, we performed chassis dynamometer experiments to investigate the emissions and secondary organic aerosol (SOA) formation potential of intermediate volatility organic compounds (IVOCs) from an on-road Chinese gasoline vehicle. High IVOCs emission factors (EFs) and distinct volatility distribution were recognized. The IVOCs EFs for the China V vehicle ranged from 12.1 to 226.3 mg kg-fuel$^{-1}$, with a median value of 83.7 mg kg-fuel$^{-1}$, which was higher than that from US vehicles. Besides, large discrepancy in volatility distribution and chemical composition of IVOCs from Chinese gasoline vehicle exhaust was discovered, with larger contributions of $B_{14}$-$B_{16}$ compounds (retention time bins corresponding to $C_{14}$-$C_{16}$ $n$-alkanes) and higher percentage of $n$-alkanes. Further we investigated the possible reasons that influence the IVOCs EFs and volatility distribution and found that fuel type, starting mode, operating cycles and acceleration rates did have an impact on the IVOCs EF. When using E10 (ethanol volume ratio of 10%, v/v) as fuel, the IVOCs EF of the tested vehicle was lower than that using commercial China standard V fuel. The average IVOC-to-THC ratios for gasoline-fueled and E10-fueled gasoline vehicle were $0.07 \pm 0.01$ and $0.11 \pm 0.02$, respectively. Cold-start operation had higher IVOCs EF than hot-start operation. Chinese Light vehicles Test Cycle (CLTC) produced 70% higher IVOCs than those from the World-wide harmonized Light-duty Test Cycle (WLTC). We found that the tested vehicle emitted more IVOCs at lower acceleration rates, which leads to high EFs under CLTC. The only factor that may influence the volatility distribution and compound composition is the engine-aftertreatment system, which has compound and volatility selectivity in exhaust purification. These distinct characteristics in EFs and volatility may result in higher SOA formation potential in China. Using published yield data and surrogate equivalent method, we estimated SOA formation under different OA loading and $NO_x$ conditions. Results showed that under low and high $NO_x$ conditions at different OA loadings, IVOCs contributed more than 80% of the predicted SOA. Furthermore, we built up a parameterization method to simply estimate the vehicular SOA based on our bottom-up measurement of VOCs and IVOCs, which would provide another

dimension of information when considering the vehicular contribution to the ambient
OA. Our results indicate that vehicular IVOCs contribute significantly to SOA,
implying that the importance of reducing IVOCs when making air pollution
controlling policies in urban area of China.

## 1 Introduction

Atmospheric fine particulate matter has great impacts on human health, regional air pollution, and global climate (Hallquist et al., 2009;Guo et al., 2014b). Organic aerosols are a major component of fine particulate matter. Secondary organic aerosol (SOA), formed from multiple generations of oxidation of thousands of organic gases and vapors, contribute 30% or more of organic aerosols in different areas of the world (Zhang et al., 2007). It has great impact on various other atmospheric processes, e.g. new particle formation and growth, and black carbon aging (Guo et al., 2020;Peng et al., 2016;Guo et al., 2016). Due to its complexity in sources and photochemical processes, SOA formation remains uncertain (Tang et al., 2019;Wang et al., 2020;Guo et al., 2014a).

A large discrepancy remains between modeled and measured SOA. One possible reason is missing SOA precursors. Apart from traditional SOA precursors, i.e. volatile organic compounds (VOCs), Robinson et al. (2007) proposed intermediate volatility organic compounds (IVOCs) as important contributors to SOA formation. IVOCs are less volatile than VOCs with effective saturation concentrations in the range of $10^3$ to $10^6$ μg/m$^3$(Donahue et al., 2006), roughly corresponds to the volatility range of $C_{12}$-$C_{22}$ $n$-alkanes. IVOCs exist mainly in the gas phase under typical atmospheric conditions. Previous studies demonstrate that IVOCs may be important SOA precursors both in ambient air and in typical source emissions i.e. gasoline vehicles, diesel vehicles and ship emissions (Huang et al., 2018;Zhao et al., 2016, 2015;Zhao et al., 2014;Yu et al., 2020). Recent model studies have shown that adding IVOC emissions into different models will greatly improve the SOA simulation results. For example, Giani et al. (2019) found a considerable OA enhancement in Po Valley (Northern Italy) when applying new S/IVOCs emission estimates and the new volatility distributions into CAMx, in which the improvement in SOA mainly due to the revised IVOC emissions. Huang et al. (2020) found a similar enhancement in SOA simulation for Yangtze River Delta (Southeast China) region when adding IVOC emissions into CAMx. They also show the importance of volatility distribution and emission parameterization for the model simulation. Therefore, understanding and

characterizing IVOC emissions, as well as their volatility distributions, is crucial for
improving numerical models that aim to predict OA.
China is in a high-growth stage with rapidly increasing number of on-road
vehicles (~26 fold in 25 years). This growth has created a substantial burden on air
quality and human health (Hallquist et al., 2016;Hu et al., 2015). Anthropogenic
emissions have become the major contributors to both primary and secondary
particles in megacities of China (Tang et al., 2018;Guo et al., 2012b). During the past
few years, many researchers have studied the gases and particulate matter emissions
from Chinese vehicles (Cao et al., 2016;Huang et al., 2015). However, none of these
studies have reported data on IVOCs emissions from Chinese gasoline vehicles.
Although Zhao et al. (2016) characterized IVOC emission in gasoline vehicles in the
United States, the results may not be applicable to China given differences in vehicle
technologies, operating conditions, and fuel quality. Therefore, understanding and
characterizing the IVOC emissions, as well as their volatility distributions from
Chinese vehicle, is of vital importance to understand the contribution of IVOCs to
SOA formation in China.
In this study, IVOCs emissions were measured from a China V gasoline vehicle
equipped with a direct inject (GDI) engine during chassis dynamometer testing. The
test matrix considered the influence of fuel type and operating conditions on the total
IVOC emission factors, including a newly designed cycle designed to simulate
Chinese driving conditions. All of the measurements were performed with the same
gasoline vehicle in order to consistently evaluate the effects of these different factors
on IVOC emissions. The emission factors (EFs), volatility and chemical speciation of
IVOC emissions from different conditions were investigated, and the SOA formation
potential were estimated.

**2 Materials and Methods**
**2.1 Testing Vehicle, Fuels and Test Cycles**
In this study, all measurements were performed on a vehicle chassis
dynamometer (Peng et al., 2017) using an in-use light-duty gasoline direct inject (GDI)

engine vehicle meeting the China V standard (similar to Euro 5). Tests were conducted with two fuels: commercial China Standard V gasoline and E10 fuel (10% ethanol by volume). The test cycles included the World-wide harmonized Light duty Test Cycle (WLTC), and the Chinese Light vehicles Test Cycle (CLTC). Furthermore, typical different acceleration rates were also tested. Detailed description and speed profiles of WLTC and CLTC are in Figure S1 in the supplementary information. The CLTC was specifically designed to simulate the driving patterns in Chinese cities while WLTC referred to the Euro VI standard and adopted as China VI testing protocol. Prior to tests, the tested vehicle was preconditioned with an overnight soak, without evaporative canister purge. Different acceleration rates were selected based on their frequency in both CLTC and WLTC, i.e. 1.2, 3.6 6.0 km/h/s, written as ACR1.2, ACR3.6 and ACR6.0), to investigate the effects of acceleration rates on IVOC emissions. All three acceleration "cycles" last for 600 s with a maximum velocity of 70 km/h. The acceleration driving cycles were set according to the criteria of identical cycle period and maximum velocity, and hence the mean velocity for each acceleration cycle is the same (Figure S2). We also measured IVOC emission factors (EFs) when the test vehicle was idling.

### 2.2 Sampling and Chemical Analysis

Tailpipe emissions were introduced to a constant volume sampler (CVS) that diluted the exhaust by a factor of 20 to 40. For WLTC and CLTC tests, IVOCs emissions were collected by sampling the diluted exhaust through a quartz filter followed by two tandem Tenax TA filled glass tubes (Gerstel, 6 mm OD, 4.5 mm ID glass tube filled with ~180 mg Tenax TA). Sampling tubes and transfer lines from the CVS were kept at a constant temperature ($27 \pm 2$ $^{o}$C). The flow rate for quartz filter was 10.0 L/min, and the flow rate for Tenax tube was set as 0.5 L/min. Dynamic blanks were also collected when the CVS was operated with only dilution air (no exhaust) to estimate the contribution of background organic vapors. Prior to sampling, the quartz filters were preheated to 550 °C in air for 6 h in clean aluminum foil using a muffle furnace to remove contaminations. Tenax tubes were preconditioned by using

Tube Conditioner (BCT700, BCT Technology LTD), at 300 °C for 3 h in pure nitrogen with a constant flow rate of 100 mL/min. All samples were sealed after sampling and stored in freezer at -20°C.

Quartz filters and Tenax tubes were analyzed using a gas chromatography/mass spectrometry (Agilent 6890GC/5975MS) equipped with a capillary column (Agilent HP-5MS, 30 m ×0.25 mm) coupled to a thermal desorption system (Gerstel, Baltimore, MD). The detailed method was described by Zhao et al.(2014). Prior to analysis, 5 µl of the internal standards ($d$10-acenaphthene, $d$12-chrysene, $d$4-1,4-dichlorobenzene, $d$8-naphthalene, $d$12-perylene, $d$10-phenanthrene and 7 deuterated $n$-alkanes) were injected into each adsorbent tube to track the IVOCs recovery.

For each test, particulate matter samples were also collected using independent Teflon and quartz filters. The Teflon filters were weighted using a microbalance (Toledo AX105DR, USA) after equilibration for 24 h in an environmental controlled room (temperature $20 \pm 1^{o}C$, relative humidity $40 \pm 3\%$) (Guo et al., 2010). A punch (1.45 cm$^2$) from each quartz filter was analyzed for organic carbon (OC) and elemental carbon (EC) via thermal-optical method using Sunset Laboratory-based instrument (NIOSH protocol, TOT) (Guo et al., 2012a). VOCs were sampled in SUUMA® polished stainless steel canisters and analyzed using GC-MS with a flame ionization detector. Total hydrocarbon (THC), nitrogen oxide, CO and CO$_2$ emissions under operation scenarios were measured using a Horiba OBS 2200 portable emission system.

**2.3 Quantification of IVOCs**

Twenty IVOCs compounds were quantified using authentic standards (Table S1). However, the majority of the IVOCs mass appears as a broad hump of co-eluting hydrocarbons and oxygenated organics. These compounds could not be resolved at the molecular level and were therefore classified as an unresolved complex mixture (UCM), which were grouped based on their volatilities.

The total mass of IVOCs was determined following the method of Zhao et al. (2014) (SI) . In short, the TIC of each sample was divided in to 11 retention time bins

corresponding to $C_{12}$-$C_{22}$ $n$-alkanes. The total mass in each bin was estimated using
the instrument response to the $n$-alkane in that bin. UCM was determined as the
difference between total IVOCs and speciated IVOCs in each bin. UCM was then
further classified into unspeciated branched alkanes ($b$-alkanes) and unspeciated
cyclic compounds following the approach of Zhao et al. (2016) (SI). The uncertainty
of the IVOCs could be ascribed to both sampling and analysis. The sampling
uncertainty was assumed as 10% (Huang et al., 2019). The uncertainty of using
$n$-alkanes as surrogate standards for the total IVOC mass was estimated to be less than
6.0% for alkanes and 30.6% for PAHs based on the analysis of a suite of standard
compounds (SI). Therefore, combined the above uncertainty, we consider a maximum
IVOCs mass uncertainty of 32.2% (SI).
Fuel-based IVOC emission factors (EF, mg/kg-fuel) were calculated using the
carbon-mass-balance method as following:
$$\text{EF}_{\text{IVOCs}} = \frac{[\Delta IVOC]}{[\Delta CO_2]} f_c$$
where $[\Delta$IVOC] represents the background-corrected mass concentration of
IVOCs, $[\Delta CO_2]$ is the background-corrected $CO_2$ concentration in the CVS expressed
in units of carbon mass and $f_c$ is the measured mass fraction of carbon in the gasoline

193    (0.82).


**3 Results and Discussion**
**3.1 Influence of Fuel, Starting Mode, and Operating Cycles on IVOC Emission**
**Factors**
Figure 1 depicts IVOC EFs of the tested China V gasoline vehicle and compares
them with previous studies. The IVOC EFs ranged from 12.1 to 226.3 mg kg-fuel[-1],
with a median value of 83.7 mg kg-fuel[-1]. The median IVOC value was ~3 times
higher than that of the US LEV-2 gasoline vehicles (21.9 mg kg-fuel[-1]), and one order
of magnitude lower than diesel-fueled non-road construction machinery and a
diesel-fueled large cargo vessel (971.1 and 800 mg kg-fuel[-1], respectively) (Qi et al.,
2019;Huang et al., 2018).
Figure 1 summarizes the influences of fuel type, starting mode, operating cycles
and acceleration rates on the total IVOC EFs. Various operating conditions may cause
different IVOCs emission and fuel consumption. In order to get a relative reliable
comparison, what we show here is all described in IVOCs EFs which consider both
IVOCs mass and the fuel consumption. Among all of the factors, acceleration rate has
the largest influence on the IVOC EFs. The fuel consumption at high acceleration rate
(6.0 km/h/s) would be higher than that at low acceleration rate (idling). Although not
emitted in IVOCs, the high consumption of the fuel would exist as other types of
carbon e.g. VOCs and $CO_2$ which may also have great effects on the atmosphere.
Therefore, the usage of IVOCs EFs can moderately balance the effects of the IVOCs
emission and fuel consumption and get a comprehensive comparison among different
acceleration rates. As the acceleration rate increases, the IVOC EF decreases, with the
median IVOC EF of ACR6.0 being one order of magnitude lower than that at idling.
Qi et al. (2019) and Zhao et al. (2016) report similar results for non-road construction
machinery and on-road diesel vehicles, where idling conditions emitted significantly
higher IVOCs than those under higher-speed cycles. They proposed that the higher
IVOC EFs at idling were the result of less efficient fuel combustion. An additional
factor in these tests may be the efficiency of the catalytic converter varying with
operating conditions (i.e. lower efficiency at idle operations).
When using commercial China Standard V gasoline, the median IVOC EF was
1.4 times greater than that using Ethanol gasoline, i.e. E10 (10% ethanol, v/v), with
median values of 91.5, and 67.6 mg kg-fuel$^{-1}$, respectively. The median THC EFs for
gasoline and E10 were 485 and 589 mg kg-fuel$^{-1}$, respectively, showing no significant
difference.
As expected, The IVOC EFs for cold-start tests was higher (83.7 mg kg-fuel$^{-1}$)
than those for hot-start tests (58.7 mg kg-fuel$^{-1}$). This reflects the reduced efficiency
of the catalytic converter during cold-start operation. The cold-start to hot-start IVOC
emission ratio is about 1.4, which is similar to the previous study (Zhao et al., 2016).
The median THC EFs for cold-start and hot-start tests are 556.2 and 507.8
mg kg-fuel$^{-1}$, respectively. Previous studies also show that cold starts have higher

THC EFs than hot start operation, but cold-to-hot ratios can span a wide range due to differences in operating conditions and model years (Jaworski et al., 2018;Drozd et al., 2016). The ratio is generally larger for more modern, heavily controlled vehicles (Saliba et al., 2017;May et al., 2014).

The median IVOC EF for CLTC was about 1.7 times of that for WLTC (103.5 versus 60.9 mg kg-fuel$^{-1}$). Similar results were also found for THC emission, with median THC EFs for CLTC and WLTC cycle as 617.3 and 420.3 mg kg-fuel$^{-1}$, respectively. Previous studies also show test cycles influence THC EFs. For example, Suarez-Bertoa et al. (2015) and Marotta et al. (2015) found that the NEDC cycle has higher THC EFs than WLTP or WLTC cycle. One possible explanation for the differences between the CLTC and WLTC IVOC EFs is the differences in acceleration rates. A histogram of acceleration rates of the two cycles (Figure S3) shows that CLTC has frequent low acceleration process compared to WLTC. 76.9% of the CLTC has acceleration rates ranging from -1.5 to 1.5 km/h/s versus 69.6% for the WLTC. The CLTC has no acceleration rate higher than 4 km/h/s, suggesting that the gasoline vehicles frequently drive in congested conditions in China.

The results from the acceleration rate cycles suggest that the frequent low acceleration rate in CLTC is responsible for the differences of the IVOC EF between CLTC and WLTC. The effect of acceleration on IVOC EFs is probably especially important in urban areas in China, which frequently have substantial traffic congestion. These results underscore the importance of developing cycles that simulate real-world Chinese driving condition e.g. CLTC, instead of using WLTC or other cycles to get relevant emissions data.

**3.2 Chemical Speciation of Chinese Vehicle IVOCs and the Relationships between Total IVOCs, POA and THC**

Figure 2 and S4 compare the chemical composition of IVOC emissions from the tested China V vehicle under different operating conditions. In general, IVOC chemical composition was similar across all the tests. Unspeciated IVOCs (UCM) dominates the total IVOCs mass (85.6 ±4.9%), including 65.2 ±5.2% for unspeciated

cyclic compounds and 20.4 $\pm 0.7\%$ for unspeciated $b$-alkanes. $n$-alkanes and speciated
aromatics contribute 10.9 $\pm 4.7\%$ and 3.5 $\pm 1.7\%$ of the total IVOC mass, respectively.
These results are similar to previous studies. For example, Zhao et al. (2016) found
the consistent composition of IVOC emissions across a wide set of vehicles.
Since the majority of the IVOC mass appears as UCM, the average mass spectra
provide additional insight into its composition. A similar distribution of mass
fragments was observed across all tests. Figure 2(b) shows the average IVOC mass
spectrum collected during an E10 CLTC test. Mass fragments associated with
aliphatic hydrocarbons ($m/z$ 43, 57, 71, 85) are the most abundant followed by those
associated with aromatics ($m/z$ 91, 105 and 119 for alkylbenzenes (Pretsch et al.,
2013), and $m/z$ 115, 165, 189 for poly aromatic species) (Dall'Osto et al.,
2009;Spencer et al., 2006).
Figures 2(c) and (d) exhibit the contribution of selected mass fragments in low
and high volatility ranges, i.e. $B_{12}$-$B_{16}$ and $B_{17}$-$B_{22}$. Aliphatic fragments are higher than
aromatics fragments in both $B_{12}$-$B_{16}$ and $B_{17}$-$B_{22}$ bins. Compared to the higher
volatility ($B_{12}$-$B_{16}$) bins, the ratio of selected aromatic to aliphatic fragments is lower
in the lower volatility ($B_{17}$-$B_{22}$) bins (0.8 versus 1.7) which suggests different
weighting of compounds in different volatility range. Therefore, unspeciated IVOC
UCM in $B_{12}$-$B_{16}$ are predominantly aromatics while $B_{17}$-$B_{22}$ are more abundant in
cyclic alkanes.
Figure 3 and S5 shows the volatility distribution of IVOC emissions over the 11
retention-time bins ($B_{12}$-$B_{22}$). IVOC emissions are more heavily weighted towards the
more volatile end of the distribution, with more than 50% of the emissions in $B_{12}$-$B_{14}$
bins. After $B_{14}$, the IVOC emission decreases significantly.
Although the IVOC EFs varied by an order of magnitude across the set of tests
(Figure 1), the volatility distributions of the emissions were largely the same. When
the vehicle is fueled by gasoline, the median IVOC fractions in the $B_{12}$-$B_{14}$ bins are
slightly higher than when fueled by E10 (Figure S5a). Cold-start has a higher median
percentage of IVOC in $B_{12}$-$B_{14}$ bins compared to hot-start (Figure S5b). No distinct
differences in volatility differences between the CLTC and WLTC (Figure S5c).

Compared with idling condition, acceleration cycles have higher median percentage of IVOC in lower volatility bins ($B_{17}$-$B_{22}$) (Figure S5d), similar to previous studies (Qi et al., 2019;Cross et al., 2015).The modest variations of volatility distributions of the IVOCs emissions may be due to differences in combustion efficiency and/or catalytic converter efficiency as a function of volatility.

Considering the similarity of volatility distribution for different conditions and the importance of the volatility distribution in model input for SOA simulation, figure S6 and Table S3 present the volatility distribution of SVOC and IVOC emissions from the tested China V gasoline vehicle, using effective saturation concentration (C*) as classification: IVOCs (C*=300-3 $\times$ $10^6$ μg·m$^{-3}$), SVOCs (C*=0.3-300 μg·m$^{-3}$ ). IVOCs are the dominant part of the low volatility organics (IVOCs+SVOCs), with a median contribution of ~95%.

Previous studies have used different scaling approaches to estimate IVOC emissions using other primary emission data, e.g. POA, NMHC (Murphy et al., 2017;Woody et al., 2016;Koo et al., 2014). However, these ratios depend on fuel, engine technology and operating conditions (Lu et al., 2018).Therefore, it is important to quantify the relationships between IVOCs and other pollutants using data collected from Chinese vehicles. Our results show that the IVOC-to-THC ratio does depend on fuel composition. The average IVOC-to-THC ratios for gasoline-fueled and E10-fueled gasoline vehicle are 0.07 $\pm$ 0.01($R^2$ = 0.87) and 0.11 $\pm$ 0.02 ($R^2$ = 0.78), respectively (Figure S7). The IVOC-to-THC ratios in this study are higher than US vehicles (IVOC-to-NMHC of 0.04) (Zhao et al., 2016) but much lower than diesel fueled vehicles (IVOC-to-THC of 0.67) (Huang et al., 2018). The IVOC-to-POA ratio was 5.12 $\pm$ 1.30 across all tests, but with only modest correlation ($R^2$ of 0.66 for gasoline-fueled vehicle and 0.43 for E10-fueled vehicle). This ratio is similar to US data for gasoline vehicles. The correlation of IVOC to THC or POA in our dataset is lower than that of the on-road gasoline and diesel vehicles measured in US. This may be caused to the US data are from a large fleet of vehicles while our data is from a single vehicle operated over a range of conditions

**3.3 High Emission Factors and Distinct Volatility Distributions of IVOCs from Chinese Gasoline Vehicles**

Figure 4 presents PM, $NO_x$, THC and IVOC EFs of the tested gasoline vehicle (China V) and compares them to US vehicles tested by Zhao et al. (Zhao et al., 2016;May et al., 2014) For this comparison, we combined all of the CLTC and WLTC data together. The US vehicles are grouped by model year where pre-LEV refers to vehicles manufactured prior to 1994, LEV-1 represents vehicles manufactured between 1994-2003, and LEV-2 indicates vehicles manufactured between 2004-2012.

The emissions of $NO_x$ and THC from tested vehicle are comparable with those from the newer (LEV-2) US vehicles tested by May et al. (Zhao et al., 2016;May et al., 2014). However, PM EF (44.8 mg kg-fuel$^{-1}$) of the tested vehicle is higher than the LEV-2 vehicles tested (17.0 mg kg-fuel$^{-1}$). It is comparable to a pre-LEV vehicle (61.0 mg kg-fuel$^{-1}$). In addition, we compared our results with that from European vehicles, and found that the $NO_x$ and THC EFs for the tested vehicle were lower than Euro 5 gasoline vehicle, while the PM EF was higher (Fontaras et al., 2014). This suggests that compared with US and European vehicles, the stringent emission implemented by Chinese government have been effective at controlling $NO_x$ and THC, but might be inefficient to PM emissions. For past 30 years, Chinese government has adopted a series of emission control policies and measures for light-duty vehicles, including implementation of emission standards for new vehicles promotion of sustainable transportation and alternative fuel vehicles, and traffic management programs (Wu et al., 2017;Zhang et al., 2014). Wu et al. (2017) summarizes the implementation of the vehicle control policies in China, which shows the control for the vehicular pollutants is becoming stricter step by step. For example, the $NO_x$ emission standard changed from 0.15 g km$^{-1}$ to 0.035 g km$^{-1}$ while the standard changed from China III to China VI. Different from $NO_x$ and THC which has been controlled since China III, only when in 2017, China V standard first introduced the control of PM into the emission control scope. Yang et al. (2020) investigated the effects of gasoline upgrade policy on migrating the PM pollution in China and found that there's no much space for significantly reducing the PM concentration by simply

improving the gasoline quality. Therefore, for PM control, more policies i.e.
developing cleaner alternatives to fossil fuels, replacing traditional vehicles with
new-energy and building developed public transport system should be done.
The IVOC EFs for the tested China V vehicle is between the US Pre-LEV and
LEV-1 vehicle. Therefore, Chinese regulations may also appear to be ineffective at
controlling IVOC emissions. The IVOC-to-THC ratio measured here (0.07 for
gasoline and 0.11 for E10) is higher than US vehicles (0.04), which means that IVOCs
contribute a larger fraction of the THC emissions from the China V than from the US
vehicles. A detailed comparison of the individual VOC emissions between China V
and US LEV-2 vehicles is in SI (Figure S9).
UCM accounts for large fraction of IVOCs for both China V and US gasoline
vehicles. However, the speciated compounds exhibit different characteristics. The
China V exhaust has less speciated IVOC aromatic compounds (3.5%) and more
alkanes (10.9%) compared to US exhaust (12.9% and 2.5%, respectively). This is also
reflected by the IVOC mass spectrum, where Chinese vehicle exhaust has higher $m/z$
43, 57, 71, 85 signals. In addition, the specific aromatics mass fragments were not the
same for China V and US IVOC emissions. For example, the dominant aromatics
fragments in US gasoline exhaust are $m/z$ 128, 119, 105, 133 versus $m/z$ 135, 91, 181,
189 for China V. (Fig. 2c and d).
Figure 3 compares the volatility distribution of the IVOC emissions from the
China V and US vehicles. There are significant differences of volatility distribution
between China V and US vehicles. Both distributions decrease with the increase of
the retention time, but the IVOC volatility distribution of US vehicle exhaust exhibits
heavier weight of lower volatility bin, i.e. $B_{12}$ bin compared to the China V vehicle. In
US exhaust the $B_{12}$ fraction is more than double of the $B_{13}$. However, the contributions
of $B_{12}$-$B_{14}$ bin volatility bins are comparable for Chinese vehicle exhaust. US vehicle
exhaust has a similar IVOC volatility distribution as the unburned gasoline, indicating
that the evaporate of IVOCs from fuel is non neglectable.
The differences between the IVOC volatility distribution between the Chinese
vehicle exhaust and unburned gasoline were further investigated. The higher emission
factor and broader distribution of IVOCs in exhaust from China V compared with US
vehicles may be due to differences in fuel composition, operating conditions and
engines and after-treatment technology. As the tests of US vehicles were all
performed using California commercial fuel, which is, in fact, E10 fuel. Therefore, in
this study, the US (unburned) fuel or US gasoline means E10. Lu et al. (2018)
demonstrated that IVOC emissions depend strongly on fuel composition. In our study,
IVOCs contributed ~2.0 wt% (2.1 wt% for gasoline, 1.9 wt% for E10) of the total fuel
mass, which is ~60% higher than the California fuel (E10, 1.2 wt%) (Gentner et al.,
2012). Therefore, the higher IVOC fractions in China V exhaust (e.g. IVOC-to-THC
ratio of 0.07 and 0.11 versus 0.04 in US exhaust) may lead to higher amounts of
IVOCs in China V gasoline. When considering volatility distribution, Zhao et al.
(2016) and Lu et al. (2018) reported similar distributions of IVOC between gasoline
vehicle exhaust and unburned fuel, which demonstrates the significant influence of
unburned fuel on exhaust volatility distribution. As a result, in Figure 3, we use US
gasoline vehicle exhaust to both represent the exhaust and the unburned (E10) fuel
and compare the Chinese E10 fuel with US fuel to get a comparative study. However,
the volatlity distribution of the China V gasoline vehicle exhaust are different from
that of the unburned fuel (Figure 3). The difference might be related to the operating
conditions and engine-aftertreatment system.
Although operating conditions strongly influence the total IVOC EFs (Figure 1),
Figure 3 indicates the volatility distribution of the IVOCs emissions were largely
consistent across the set of test conditions. Therefore, operating conditions cannot
explain the difference in the IVOC volatility distribution between the China V vehicle,
unburned gasoline, and the US vehicles.
The engine-aftertreatment system also influences IVOC emissions (Drozd et al.,
2019;Alam et al., 2019;Zhao et al., 2018;Saliba et al., 2017). In order to investigate
the efficiency of after-treatment system, we normalized the IVOC distributions of the
fuel and exhaust to the sum of $C_8$-$C_{10}$ $n$-alkanes. It is believed that the $C_8$-$C_{10}$
$n$-alkanes can serve as the indicators for VOCs in fuel (Lu et al. 2018). For both US
and the China V vehicles, IVOCs are enriched in the exhaust relative to the fuel.
However, the enrichment factor is much smaller in Chinese exhaust with a median
value of 4.0 than that for US vehicles (median value = 8.5) (Lu et al., 2018). The
enrichment factor also varies with different compounds, with the enrichment factors
of *n*-alkanes (9.3)＞*b*-alkanes (6.6)＞unspeciated cyclic ompounds (3.1)＞aromatics
(0.4). These results are consistent with previous studies stating that the aftertreatment
devices have different removal efficiency towards different compounds (Ma et al.,
2019;Hasan et al., 2018;Hasan et al., 2016;Alam and Harrison, 2016). Our results
suggest that the Chinese three-way catalytic converter has compound dependent
efficiency (better removal of aromatics compared to alkanes) which might explain the
difference in compound composition between Chinese and US vehicle exhaust.
Furthermore, Fig. S10 shows that the catalytic converter has different removal
capacity towards different volatility bins, in which $B_{14\text{-}16}$ works much worse compared
to other volatility bins i.e. $B_{12}$. Consequently, the SOA formation would be relatively
high. In sum, the compound dependent capacity and lower $B_{14}$-$B_{16}$ removal efficiency
of Chinese TWC is responsible for the volatility distribution differences between
China V and US vehicles shown in Figure 3.
After considering all the factors above, we can draw the conclusion that fuel type,
starting mode and operating conditions can all affect the IVOCs EFs. The only factor
that impacts the volatility distribution is engine-aftertreament system.
**3.4 Estimation of SOA Production from Chinses Vehicle Emission**
With the measured IVOC and VOC emissions, we estimated the SOA formation
potential by using the yield method as following (Yuan et al., 2013):
$$\Delta\text{SOA}/\Delta\text{CO} = \sum ER_{[HC]_i} \times \left(1 - e^{-\left(k_{\text{OH,i}} - k_{\text{CO}}\right) \times [OH] \times \Delta t}\right) \times Y_i$$
In which, $ER_{[HC]_i}$ is the emission ratio of SOA precursor i (mg kg-fuel$^{-1}$); $k_{OH,i}$
is the OH reaction rate constant of precursor i at 298K (cm$^3$ molecules$^{-1}$ s$^{-1}$); $k_{\text{CO}}$ is
the OH reaction constant of CO at 298 K (2.4×10$^{-13}$ cm$^3$ molecules$^{-1}$ s$^{-1}$); [OH] is the
OH mixing ratio, which is assumed to be 1.5×10$^6$ molecules cm$^3$ (Lu et al., 2019); $\Delta t$
is photochemical age (h); and $Y_i$ is the SOA yield determined from chamber studies.
Previous studies have shown that the SOA yield of individual hydrocarbon can be

influenced by $NO_x$ level, due to the competition reactions among $RO_2$ radicals, NO and $HO_2$ radicals. Usually SOA yields under low $NO_x$ condition are independent on the OA loading. However, under high $NO_x$ condition, SOA yields highly depend on OA mass concentration, which can be described using two-product or multi-products model (Presto et al., 2010;Chan et al., 2009;Ng et al., 2007). In this study, we estimated SOA formation under low and high $NO_x$ conditions with OA concentration of 10, 20, 80 $\mu g \cdot m^{-3}$ to represent the influence of $NO_x$ level and OA loading on SOA formation.

In this estimation, we include speciated $C_6$-$C_9$ single ring aromatics (SRAs) as typical VOCs for SOA precursors, and the corresponding $k_{OH}$ and SOA yields are extrapolated according to two-product relationship from chamber studies (see SI) (Ng et al., 2007). The SOA yields under low and high $NO_x$ condition, and the OH reaction rates of speciated IVOCs and SRAs are from the previous studies (see SI) (Presto et al., 2010;Lim and Ziemann, 2009;Chan et al., 2009). In brief, surrogate species were used to represent the unspeciated *b*-alkanes and cyclic compounds in each of the volatility bins.

Figure 5 shows the POA emission and estimated SOA production under different operating conditions and $NO_x$ level after 48 h of photo-oxidation. The estimated SOA/POA ratio is between 4.0 to 5.0 under low $NO_x$ condition, and the SOA-to-POA ratios ranged from 1.8-2.2 to 3.8-4.4 when the OA loading increased from 10 $\mu g \cdot m^{-3}$ to 80 $\mu g \cdot m^{-3}$ under high $NO_x$ condition. The OA enhancement under low $NO_x$ condition is similar to that under high $NO_x$ condition with the OA loading of 80 $\mu g \cdot m^{-3}$. Considering the high POA concentration and SOA formation capacity of Chinese gasoline vehicles, the SOA/POA ratios at 80 $\mu g \cdot m^{-3}$ are considered as a lower estimation. Compared with OA enhancement from US studies (~3.6) (Zhao et al., 2016), our results showed higher SOA formation potential both under low and high $NO_x$ conditions for Chinese gasoline vehicles.

Scenario-based analysis shows similar tendency of SOA formation potential at different OA loading under low and high $NO_x$ condition. Though the POA emission for gasoline-fueled vehicle was higher than that fueled by E10, comparable SOA

formation is estimated using gasoline and E10 as fuel. That means, the OA enhancement factor for E10 is higher than that of gasoline. This suggests that although the ongoing policy of ethanol gasoline will not exacerbate the POA emission in China, the SOA formation of E10 could not be neglected due to its high SOA enhancement capacity. Therefore, more research should be done to evaluate the effectiveness of using E10 as surrogate to reduce the air pollution in China.

Cold-start operation has higher SOA potential with higher OA enhancement factor than hot-start, due to the higher precursors EFs caused by the reduced catalytic converter effectiveness below its light off temperature (Drozd et al., 2019). The IVOC EFs, the estimated SOA production and SOA/POA of CLTC are all higher than those of WLTC, which further demonstrates the higher SOA formation potential of Chinese gasoline vehicles under typical driving conditions in China.

Figure S11 presents the contribution of different classes of precursors on the SOA production after 48 h of photo-oxidation under different OA loading and $NO_x$ condition. The relative contributions of different chemical classes were similar across the different conditions, with the largest contribution from unspeciated cyclic IVOCs. This is different from the US gasoline vehicle SOA (Zhao et al., 2016) in which single ring aromatics contributes the most.

**3.5 Establishing the Estimation Method of SOA formation from Chinese Gasoline Vehicles**

In this section, we tried to establish parameterization methods to provide simple estimations of gasoline vehicle SOA based on our measurements of VOCs and IVOCs.

Figure S12 shows the average predicted SOA-to-POA ratio as the function of photo-oxidation time under different OA loading and $NO_x$ conditions. In general, SOA exceeds POA after first a few hours of oxidation, and then levels off after 30 h. The SOA/POA ratio is influenced by OA concentration, $NO_x$ level and the photochemical age (OH exposure). At a certain OA loading and OH exposure, SOA/POA ratio can be estimated, and then be used to quantify the contributions of

gasoline vehicle SOA to the ambient OA. Therefore, we parameterized the SOA/POA variation under different OA and $NO_x$ condition using three-parameter-based logarithm equation, i.e. $y = a - b \times \ln(t+c)$, in which $t$ represents the equivalent photochemical age (assume that the OH concentration is $1.5 \times 10^6$ molecules cm$^3$) and a, b, c can be described using three-parameter logarithm equation $y = m - n \times \ln([\text{OA concentration}] + p)$. Table 1 shows the parameterization results of compounds-based SOA/POA variation under the different OA and $NO_x$ condition. The fits quality could be found in Figure S13.

Table 1 Coefficient of parameterization between SOA/POA and photochemical age

| SOA/POA | Low NO$_x$ condition | High NO$_x$ condition | | |
|---|---|---|---|---|
| | | m | n | p |
| a | -0.62 | 0.46 | 0.22 | 9.8 |
| b | -1.34 | 0.27 | 0.33 | 2.58 |
| c | 0.58 | 0.13 | -0.09 | 3.35 |
| Unspeciated cyclic compounds | | | | |
| a | -0.15 | 0.26 | 0.09 | 21.76 |
| b | -0.72 | 0.086 | 0.18 | 0.46 |
| c | 0.11 | -0.278 | -0.083 | 24.42 |
| Unspeciated b-alkanes | | | | |
| a | -0.11 | 0.47 | 0.111 | 87.54 |
| b | -0.17 | 0.15 | 0.070 | 12.36 |
| c | 0.84 | -0.17 | -0.21 | 41.97 |
| aromatics | | | | |
| a | -0.03 | -0.023 | -0.0098 | 40.52 |
| b | -0.03 | 0.012 | 0.007 | 17.27 |
| c | -1.00 | -1.02 | -0.021 | -10.00 |
| n-alkanes | | | | |
| a | -0.05 | 0.0067 | 0.013 | -2.38 |
| b | -0.11 | 0.019 | 0.030 | -0.52 |
| c | 0.48 | 0.15 | -0.058 | 29.18 |
| Single          ring | | | | |

| aromatics | | | | |
|---|---|---|---|---|
| a | -0.51 | 0.28 | 0.17 | 5.47 |
| b | -0.35 | 0.03 | 0.059 | -2.29 |
| c | 3.92 | 2.80 | -1.29 | 10.84 |

The above photochemical-based parameterization method provides a conservative
way to quantify the evolution of SOA from Chinese gasoline vehicle VOCs and
IVOCs oxidation. However, there are still some uncertainties which may lead to
discrepancies between predicted and measured SOA. In general, positive or negative
artifacts of quartz filters, $n$-alkane equivalent method in estimating the IVOCs
concentration, uncertainty in SOA yield, surrogate method to substitute SOA yield
and $k_{OH}$ for UCM and lack of semi-volatile organic compounds will exert influence on
the SOA prediction.

**4 Atmospheric Implications**
We measured the VOCs, IVOCs and POA emitted from a China V light duty
gasoline vehicle across a wide range of operating conditions. Compared with US
LEV-2 gasoline vehicles, the China V vehicle emits three times more IVOCs. Besides,
the IVOC emissions from the China V vehicle have a much broader volatility
distribution than that from US vehicles. These characteristics imply that IVOCs could
act more important SOA precursors in China than those in the US. For Chinese
gasoline vehicles, although the magnitude of the emission of IVOCs and VOCs can
vary, their relative contribution to SOA production is similar across the set of
operating conditions examined here due to the similar volatility distributions. As a
result, the key to control SOA formation of gasoline vehicles is to reduce the total
IVOC EFs by upgrading of emissions controls. In addition, reducing congestion and
other low speed operating modes would also be effective at reducing emissions
(Figure 1 and 5).
Based on our results, we roughly estimate the vehicle IVOC emissions in China.
Till the end of 2018, the total vehicle population in China reached 0.327 billion, with
automobiles contributing 61% (0.24 billion). Of all the automobiles, gasoline-fueled

car took the dominant (88.1%). The HC emission of gasoline vehicles in China was 0.23 Mt, accounting for more than 70% of the total vehicle emissions. Using an IVOC/THC ratio of 0.09 that is obtained in our work, we estimate that the vehicle IVOC emissions in China are 0.03 Mt (30 Gg), in which 20 Gg is attributed to gasoline vehicles. One should note that this estimation is a conservative value, since we consider all vehicles as gasoline vehicles, and all of them meet the China V standard. According to the statistics from the Ministry of Ecology and Environment, only 30.9% of the vehicles in 2018 meet the standards of China V. Indeed, higher percentage of pre-China V e.g. China I-IV standard cars will cause more IVOCs emission. In addition, the IVOC/NMHC ratio of diesel vehicles could be much higher than that of the gasoline vehicles (Zhao et al., 2016, 2015). This may also lead to an underestimation.

Our results show that using a Chinese real-world test protocol CLTC will result in substantially higher IVOC emissions compared with WLTC which might have close relationship with frequent idling and low acceleration condition. Therefore, when driving at typical Chinese condition where traffic congestion frequently occur, the IVOCs emission from Chinese gasoline vehicles would be much higher than the current limited emission inventory. Our results indicate simply controlling the THC, $NO_x$ and primary PM emissions may be insufficient in the aspect of controlling particle pollution. Reducing IVOC emissions should also be taken into consideration due to their high contribution to SOA formation, which is more important than primary organic aerosol. Suggested controlling ways include upgrading the fuel quality and engine-after treatment system, and reducing the traffic congestion.

Though we have discussed the influences of different operating conditions on IVOC emissions and SOA formation for the tested China V gasoline vehicle, due to the singular vehicle tests of our study, more research i.e. vehicles meeting different emission standards, different engines should be performed both to testify the accuracy of our research and to get a full understanding of the IVOC emission inventory for Chinese gasoline vehicles. Furthermore, advanced measurement techniques e.g. GC×GC-MS and chemical ionization mass spectrometry (CIMS) should be used to obtain

a comprehensive molecular-level picture of the total organics so as to reduce the
uncertainties associated with the measurements and models.

*Data availability*
The data used in this publication are available on
https://doi.org/10.5281/zenodo.4072847, and they can be accessed by request to the
corresponding author (songguo@pku.edu.cn) of Peking University. The IVOCs
emissions from US vehicles used in this study can be accessed via
https://pubs.acs.org/doi/abs/10.1021/acs.est.5b06247 (Zhao et al., 2016) and
https://acp.copernicus.org/articles/18/17637/2018/ (Lu et al, 2018). The primary
emissions and fuel compositions from US vehicles used in this study can be accessed
via https://www.sciencedirect.com/science/article/pii/S1352231014000715 (May et al.,

2014).



*Author contributions*
SG, RZ and HW designed the study. RZ and KS collected the samples. RZ and QL
analyzed the samples and processed the data. RZ wrote the paper, with contributions
from all the coauthors.

*Acknowledgement*
This research is supported by the National Key Research and Development Program
of China (2016YFC0202000), the National Natural Science Foundation of China (No.
51636003,41977179,21677002,91844301), Beijing Municipal Science and
Technology Commission (Z201100008220011), and Natural Science Foundation of
Beijing (No. 8192022), the Open Research Fund of State Key Laboratory of
Multi-phase Complex Systems (MPCS-2019-D-09). ALR and QY received financial
support from the Center for Air, Climate, and Energy Solutions (CACES), which was
funded by Assistance Agreement No. RD83587301 awarded by the U.S.
Environmental Protection Agency. It has not been formally reviewed by EPA. The

views expressed in this document are solely those of authors and do not necessarily reflect those of the Agency. EPA does not endorse any products or commercial services mentioned in this publication.

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

 **Figure Caption**

**Figure 1.** IVOC emission factors measured under different conditions, i.e. different
fuel type (gasoline, E10), test cycles (Chinese Light vehicles Test Cycle, CLTC,
and World-wide harmonized Light-duty Test Cycle, WLTC), starting mode (hot
start and cold start), and acceleration rates (acceleration rates of 1.2, 3.6 and 6.0
km/h/s). Stars indicate the EF data from US, i.e. median US LEV-2 gasoline
vehicles (vehicles manufactured in 2004-2012), non-road construction machinery,
and a large cargo vessel (Qi et al., 2019;Huang et al., 2018;Zhao et al., 2016). The
first category "China V" is the compilation of all the EF results from all of the
CLTC and WLTC tests. The boxes indicate the median value, with error bars
represent one standard deviation.
**Figure 2.** (a) Comparison of average chemical speciation of IVOC emissions from
China V vehicle and US vehicles (Zhao et al., 2016); (b) Average mass spectrum
of the IVOC during a typical E10-fueled cold start CLTC test. (c-d) Box-whisker
plots of the fractional contribution of selected fragments to total IVOCs signal for
tested China V vehicle:(c) $B_{12}$-$B_{16}$ bins; (d) $B_{17}$-$B_{22}$ bins. The boxes represent the
$25^{th}$ and $75^{th}$ percentiles with the centerline being the median. The whiskers are the
$10^{th}$ and $90^{th}$ percentiles. Black hollow triangles represent median LEV-2 data
from Zhao et al. [13] LEV-2 represents vehicles manufactured from 2004 to 2012.
Fragments colored in blue represent aliphatic compounds, while those colored in
orange are associated with aromatic compounds.
**Figure 3.** Comparison of IVOC volatility distributions of Chinses gasoline vehicle
exhaust, US gasoline vehicle exhaust, and Chinses E10 fuel. The box-plot
represents the Chinses gasoline vehicle exhaust. The boxes represent the $25^{th}$ and
$75^{th}$ percentiles with the centerline being the median. The whiskers are the $10^{th}$ and
$90^{th}$ percentiles. Red solid circles represent IVOC fractions of US vehicle exhaust
(Zhao et al., 2016). Blue hollow triangles represent the IVOCs volatility
distribution of Chinese E10 fuel. As all the studies performed in US used
commercial US gasoline as fuel, which contained 10% v/v ethanol, i.e. E10 fuel.
Therefore, we compare the Chinese E10 with US fuel to get a consistent
comparison. Also, we should note that Zhao et al. (2016) and Lu et al. (2018)
found that consistent distribution of US fuel and exhaust, so in this figure, the US
gasoline vehicle exhaust can represent the volatility distribution of its unburned
fuel distribution as well.
**Figure 4.** Comparison of emission factors of (a) PM (b) $NO_x$ (c) THC, and (d) IVOC
between China and US on road gasoline vehicles (Zhao et al., 2016;May et al.,
2014). The boxes present the $75^{th}$ and $25^{th}$ percentiles with the centerline
represents being the median. The US vehicles are grouped by the model year, i.e.
pre-LEV refers to vehicles manufactured prior to 1994, LEV-1 represents vehicles
from 1994-2003, and LEV-2 is vehicles manufactured from 2004-2012.
**Figure 5** Comparison of POA and estimated SOA production after 48 h of
photo-oxidation (a) under low $NO_x$ condition; (b) at an OA loading of 10 $\mu g \cdot m^{-3}$
under high $NO_x$ condition; (c) at an OA loading of 20 $\mu g \cdot m^{-3}$ under high $NO_x$
condition; (d) at an OA loading of 80 $\mu g \cdot m^{-3}$ under high $NO_x$ condition. The blue
circles represent the SOA-to-POA ratio after 48 h of photooxidation (right axis).

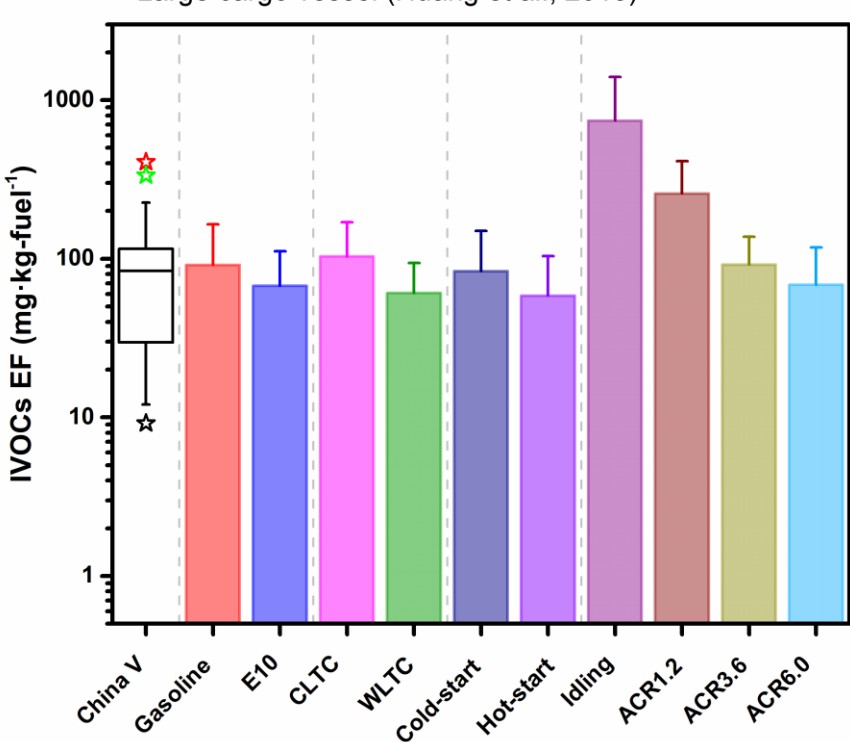


**Figure 1.** IVOC emission factors measured under different conditions, i.e. different fuel type (gasoline, E10), test cycles (Chinese Light vehicles Test Cycle, CLTC, and World-wide harmonized Light-duty Test Cycle, WLTC), starting mode (hot start and cold start), and acceleration rates (acceleration rates of 1.2, 3.6 and 6.0 km/h/s). Stars indicate the EF data from US, i.e. median US LEV-2 gasoline vehicles (vehicles manufactured in 2004-2012), non-road construction machinery, and a large cargo vessel (Qi et al., 2019;Huang et al., 2018;Zhao et al., 2016). The first category "China V" is the compilation of all the EF results from all of the CLTC and WLTC tests. The boxes indicate the median value, with error bars represent one standard deviation.

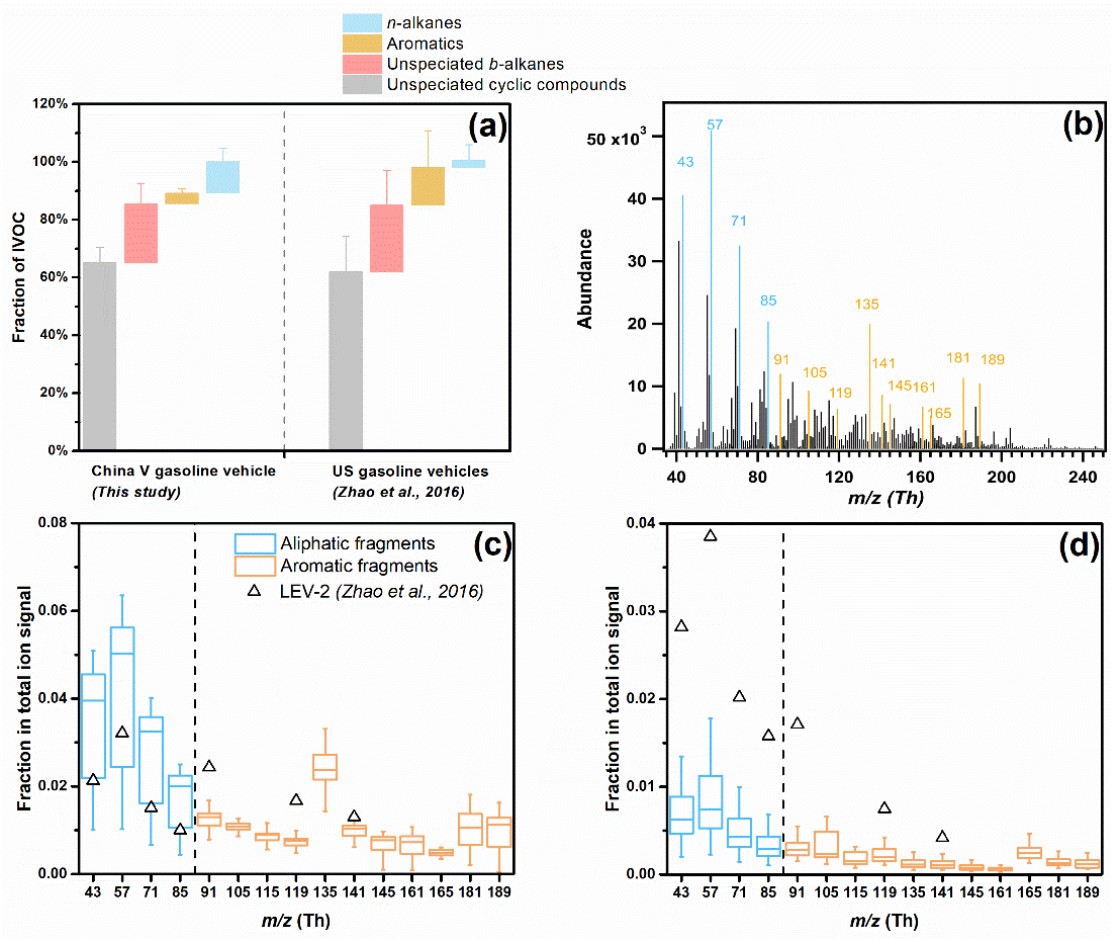


**Figure 2.** (a) Comparison of average chemical speciation of IVOC emissions from
China V vehicle and US vehicles (Zhao et al., 2016); (b) Average mass spectrum of
the IVOC during a typical E10-fueled cold start CLTC test. (c-d) Box-whisker plots of
the fractional contribution of selected fragments to total IVOCs signal for tested
China V vehicle:(c) $B_{12}$-$B_{16}$ bins; (d) $B_{17}$-$B_{22}$ bins. The boxes represent the 25[th] and
75[th] percentiles with the centerline being the median. The whiskers are the 10[th] and
90[th] percentiles. Black hollow triangles represent median LEV-2 data from Zhao et al.
(2016) LEV-2 represents vehicles manufactured from 2004 to 2012. Fragments
colored in blue represent aliphatic compounds, while those colored in orange are
associated with aromatic compounds.

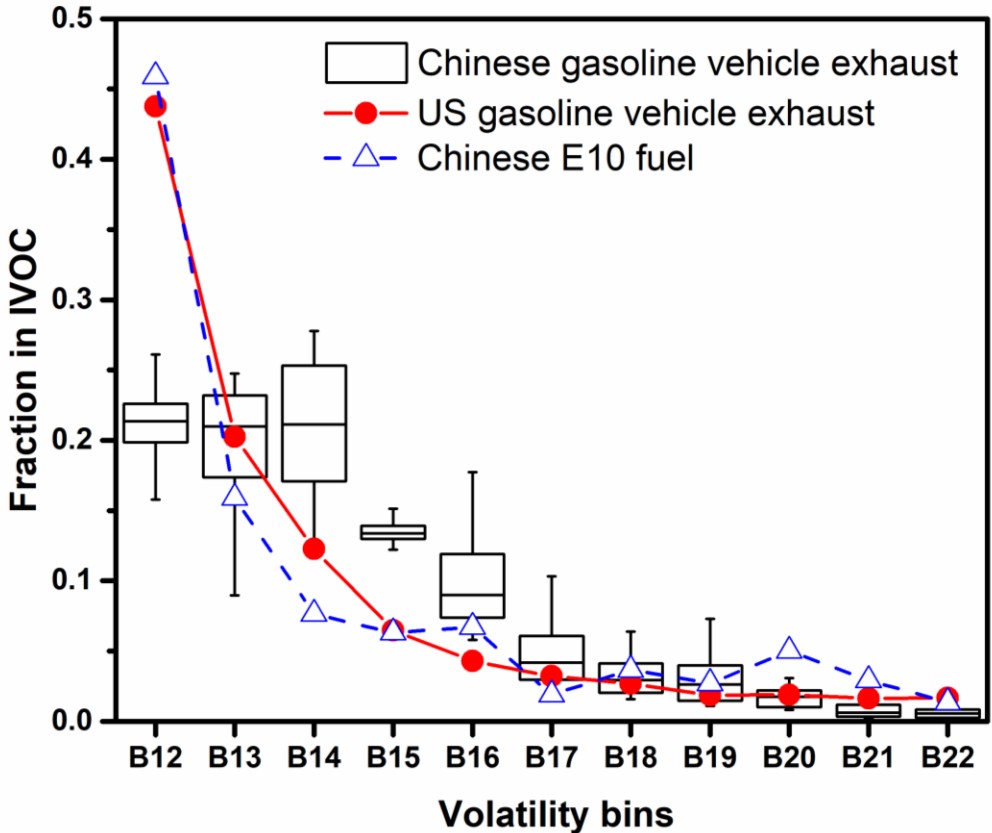


**Figure 3.** Comparison of IVOC volatility distributions of Chinses gasoline vehicle

exhaust, US gasoline vehicle exhaust, and Chinses E10 fuel (ethanol volume ratio of

10%, v/v). The box-plot represents the Chinses gasoline vehicle exhaust. The boxes

represent the $25^{th}$ and $75^{th}$ percentiles with the centerline being the median. The

whiskers are the $10^{th}$ and $90^{th}$ percentiles. Red solid circles represent IVOC fractions

of US vehicle exhaust (Zhao et al., 2016). Blue hollow triangles represent the IVOCs

volatility distribution of Chinese E10 fuel. As all the studies performed in US used

commercial US gasoline as fuel, which contained 10% v/v ethanol, i.e. E10 fuel.

Therefore, we compare the Chinese E10 with US fuel to get a consistent comparison.

Also, we should note that Zhao et al. (2016) and Lu et al. (2018) found that consistent

distribution of US fuel and exhaust, so in this figure, the US gasoline vehicle exhaust

can represent the volatility distribution of its unburned fuel distribution as well.

901

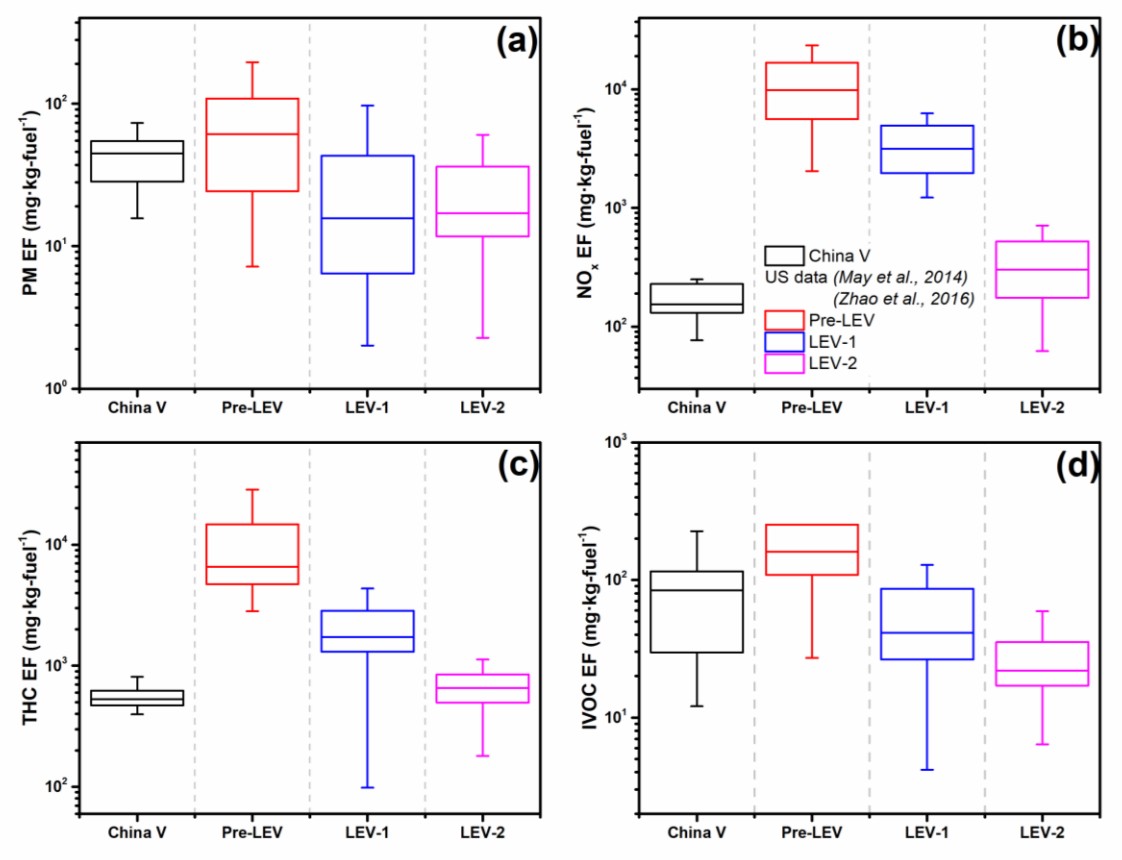

902

**Figure 4.** Comparison of emission factors of (a) PM (b) $NO_x$ (c) THC, and (d) IVOC

between China and US on road gasoline vehicles (Zhao et al., 2016;May et al., 2014).

The boxes present the $75^{th}$ and $25^{th}$ percentiles with the centerline represents being the

median. The US vehicles are grouped by the model year, i.e. pre-LEV refers to

vehicles manufactured prior to 1994, LEV-1 represents vehicles from 1994-2003, and

LEV-2 is vehicles manufactured from 2004-2012.

909

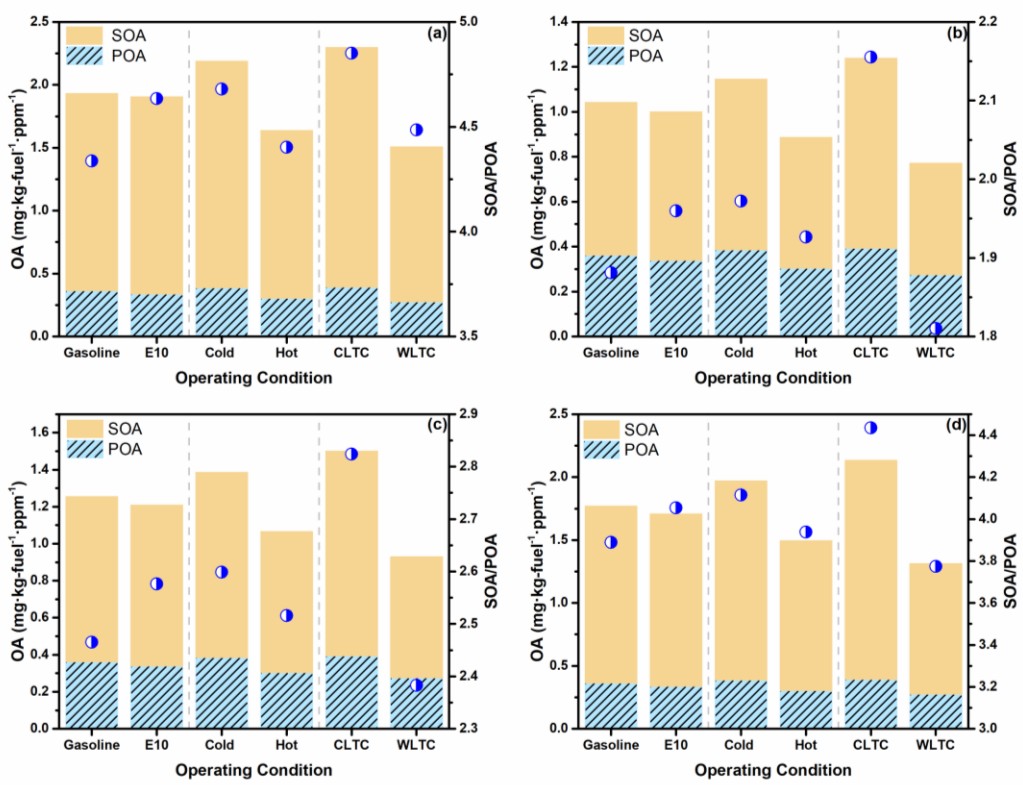

**Figure 5** Comparison of POA and estimated SOA production after 48 h of photo-oxidation (a) under low NO$_x$ condition; (b) at an OA loading of 10 μg·m$^{-3}$ under high NO$_x$ condition; (c) at an OA loading of 20 μg·m$^{-3}$ under high NO$_x$ condition; (d) at an OA loading of 80 μg·m$^{-3}$ under high NO$_x$ condition. The blue circles represent the SOA-to-POA ratio after 48 h of photooxidation (right axis).