# Peer review of "Measurement report: Distinct Emissions and Volatility Distribution of Intermediate"

_Atmospheric Chemistry and Physics, 2020_

## Referee Comment (RC1) · Anonymous Referee #1 · 7 Dec 2020

The manuscript describes a study of IVOC emission factors - determined on a test stand - exploring different driving cycles and habits and two different fuels. The authors classified the group of compounds where speciation was not possible. By using their emissions factors and compound classes they estimate the SOA formation potential and related it to real and potential measures of emission reduction. The study was performed in thorough way, the presentation of the data is very good with few exception (see comments). The paper is written well, language is good and understandable.

[Figure]

However, there are a few typos, missing articles, plural/singular issues and similar, which the author will able to lift easily. This is also true for the supplement. The obvious weak point of the paper is that the authors tested only one single car. This makes it difficult to judge in how far the results are representative at all and for the Chinese gasoline fleet. However, the authors covered a lot of different and important aspects. They also made comparisons to previous studies for US cars and driving cycle. From this point of view, I see this study as a kind of pilot study, from which a lot can be extracted for structuring future extended studies. I have only a few minor issues, and after the authors address these points, the paper can be published in ACP more or less as it is with minor changes.

Minor comments

Line 156-158, Do you have any idea how accurate that approach is? Finally, you calculate SOA potentials based on those numbers. I suggest that you extend on the potential error of the "Zhao"-method in the supplement and make a short statement here about the potential uncertainty.

Line 181-183, yes, the emission factors are lower, but the gasoline consumption is higher. Isn't it the folding of both which is important for the atmospheric effect?

Line 251 / caption Figure 3, Figure 3 needs a better introduction and captions especially introducing the Chinese E10 trace.

Line 255-265, I suggest always (4x) to refer to the panels in FigureS5 in the supplement in order to relate the statements to the plots. It is easier for the reader.

Line 296-298, these are results from only one vehicle, therefore I suggest to formulate the conclusions a bit more careful.

Line 337-340, I cannot see this in Figure 3. Or should I compare to E10 fuel? However, why E10 fuel then? As mentioned already, Figure 3 needs better explanations.

Line 410-412, I am sorry, but this sentence does not make sense to me. (grammar?)

Please, rephrase.

Line 421/Figure S10, wouldn't it be good to indicate the contribution of the classes to the emission (Figure S4). Or bring Figure S4 and S10 closer together. I guess the aromatics in Figure S4 contain also the single ring aromatics. That would mean here that aromatics are over-effective in SOA formation.

Line 434-436: "...and then keeps constant after~24 h." No, I would say it does not become constant within the first 48h. I could agree with a formulation "levels off after 30h", or "the curves flatten after 24.-30h".

Line 444 and Table 1, could you show the quality of your fits? E.g. plotted into Figure S11?

Line 758 / Figure 5, explain the "balls" in the caption

Errors:

Line 27, B14-B16 compounds, this notation cannot be used here as it is not explained.

Line 30, I suggest to use "did" have instead of "could" have

Lines 35, I would replace "vehicle" by "the tested vehicle", or so. In any case "the" is missing

Line 58 and many more instances: a blank is missing in the reference listings.

Line 104, it was only one vehicle, so please use singular

Line 120/121, either articles or use of plural for "quartz filter(s)" and "TENAX tube(s)"

Line 129, "a" gas chromatograph mass spectrometer or mass spectrometr"y"

Line135, you can skip "in the literature"

Line 144, please explain the notation SUUMA

Line 209, ... found "that the" NECD cycle..., or so

Line 251 ..."show"..."over" the 11 retention time bins.

Line 296, ...emission "measures" implemented..., or so

Line 300, Chinese regulations "may" also appear...

Line 305, Figure S8, I guess

Line 322, ..has a similar IVOC volatility distribution "as" the unburned gasoline...

Line 366 and more place, typo in the word "Chinese", please double check and correct.

Line 415, I would start a new paragraph here, beginning with "Cold start..."

Line 460, Compared with US LEV-2 gasoline vehicle"s", "the" China V vehicle emits three times "more" IVOCs. Three suggested changes in "" "".

Line 463-464, ...IVOCs could act "as" more important SOA precursors...

I found typos in the supplement (which has no line numbering and page numbers), which you can find by searching: hot-start; Zhao et al. (ref.) => Zhao et al. (2016); "b-alkane" is double.

---

## Referee Comment (RC2) · Anonymous Referee #2 · 11 Dec 2020

**General comments**

The manuscript presents novel data regarding IVOC emission factors for a gasoline/E10 Chinese vehicle, that meets China V standard. Methods are sound, the language is cogent and very easy to follow. The presentation of the results is very clear and the main findings are thoroughly discussed and compared to previous literature, considering differences and similarities with US-based data. As the paper entails im-

portant implications for both the scientific community and policymakers, I recommend final publication after minor revisions. The following comments are mostly aimed to improve the readability, interpretability and usefulness of the study for future work.

Specific comments

1. To facilitate the use of your new data in modeling studies using the Volatility Basis Set (VBS) scheme, I would recommend to present the volatility distribution data also in terms of saturation concentration bins, in a similar way to Zhao et al. [2016] (Figure 4). Also, it would be convenient if you can report a Table, maybe in the SI, reporting the mass fraction distribution of organics for each saturation concentration bin (e.g. Table S5 in Zhao et al. [2016]). These values are usually a key input for the VBS schemes in state-of-the-art numerical models. In addition to this, I would suggest to report the median IVOC-to-THC ratio in the abstract as well, as that is key information for modelers.

2. In the "atmospheric implications" section, I would suggest to at least mention the possible limitations of the study, and maybe possible future directions. One example could be the fact that only one vehicle was tested (China V), and different values might be obtained for different vehicles (even vehicles that meet the China V emission standard), implying that the total uncertainty associated with the estimated emission factors might be larger. Also, when discussing why your estimate of total IVOC emissions in China is conservative (lines 476-480), can you report what is the current percentage of vehicles that meet the China V standard in the Chinese car fleet? This would help the reader understanding the extent of the implications of the assumption made in estimating that the total IVOC emissions in China are 30 Gg.

3. In Section 3.3, you mention several times that recent Chinese regulations failed in controlling PM emissions (and IVOC emissions as well), whereas they were effective for NOx and THC, according to your data. Can you expand on that? Which regulations did they implement? Why do you think they were ineffective for PM and IVOCs but

effective for NOx and THC? Maybe some additional references might help – Expanding the discussion on this point can be useful to guide policymaking.

4. Some claims in the introduction can be better substantiated by referencing previous literature. E.g. lines 58-59 "A large discrepancy remains between modeled and measured SOA. One possible reason is missing SOA precursors." Two recent modeling works that discussed these two points are Giani et al. [2019] in Europe and [Huang et al., 2020] in China, and I suggest to add a citation to strengthen your claims. In the introduction, I would also stress the point that understanding and characterizing IVOC emissions, as well as their volatility distributions, is crucial for improving numerical models that aim to predict OA.

5. I am a little skeptical about the parametrization presented in Section 3.5, which seems somewhat arbitrary. Does the logarithmic curve have some sort of physical insight or is it based only on the shape of the calculated curve? Why not using something like k-exp(...) as in the actual model used to derive that curve (Equation in Section 3.4), also because you're claiming that after 24h SOA/POA is approximately constant? The other concern that I have is that there are a lot of parameters to be estimated (9 in high-NOx conditions), which might cause overfitting to your data, thus losing generalizability. Is it a specific reason why you're using so many parameters? Is there a way of having a simpler parametrizations with similar fit performance? If so, a simpler model (i.e. with less parameters) should be preferred. I would suggest that at least you should better justify your choices for the proposed parametrization in Section 3.5. I believe that Section 3.5 can be largely improved, either by better substantiating your choices or performing some further calculations (that might exceed the scope of the paper, though).

6. What are the dots in Figure 5? Please explain in the caption. (I'm assuming is the SOA/POA ratio to be read on the right scale?)

References

Giani, P., A. Balzarini, G. Pirovano, S. Gilardoni, M. Paglione, C. Colombi, V. L. Gianelle, C. A. Belis, V. Poluzzi, and G. Lonati (2019), Influence of semi-and intermediate-volatile organic compounds (S/IVOC) parameterizations, volatility distributions and aging schemes on organic aerosol modelling in winter conditions, Atmospheric environment, 213, 11-24.

Huang, L., Q. Wang, Y. Wang, C. Emery, A. Zhu, Y. Zhu, S. Yin, G. Yarwood, K. Zhang, and L. Li (2020), Simulation of secondary organic aerosol over the Yangtze River Delta region: The impacts from the emissions of intermediate volatility organic compounds and the SOA modeling framework, Atmospheric Environment, 118079.

Zhao, Y., N. T. Nguyen, A. A. Presto, C. J. Hennigan, A. A. May, and A. L. Robinson (2016), Intermediate volatility organic compound emissions from on-road gasoline vehicles and small off-road gasoline engines, Environmental science & technology, 50(8), 4554-4563.

---

## Author Comment (AC1) · 9 Jan 2021

We thank the referee for the careful review and suggestions. Following is our response to the comments:

➢ *Referee #1:*

*The manuscript describes a study of IVOC emission factors - determined on a test stand - exploring different driving cycles and habits and two different fuels. The authors classified the group of compounds where speciation was not possible. By using their emissions factors and compound classes they estimate the SOA formation potential and related it to real and potential measures of emission reduction. The study was performed in thorough way, the presentation of the data is very good with few exception (see comments). The paper is written well, language is good and understandable. However, there are a few typos, missing articles, plural/singular issues and similar, which the author will able to lift easily. This is also true for the supplement. The obvious weak point of the paper is that the authors tested only one single car. This makes it difficult to judge in how far the results are representative at all and for the Chinese gasoline fleet. However, the authors covered a lot of different and important aspects. They also made comparisons to previous studies for US cars and driving cycle. From this point of view, I see this study as a kind of pilot study, from which a lot can be extracted for structuring future extended studies. I*

*have only a few minor issues, and after the authors address these points, the paper can be published in ACP more or less as it is with minor changes.*

**Minor Comments**

*1. Line 156-158, Do you have any idea how accurate that approach is? Finally, you calculate SOA potentials based on those numbers. I suggest that you extend on the potential error of the "Zhao"-method in the supplement and make a short statement here about the potential uncertainty.*

Response: We thank the reviewer for the suggestion. We make a statement in the revised manuscript to indicate the uncertainties of the method in Zhao et al.

"The uncertainty of the IVOCs could be ascribed to both sampling and analysis. When sampling, the positive/negative adsorption/desorption of the target compounds on quartz filters/Tenax tubes (May et al., 2013) and slight flow fluctuation will cause sampling uncertainty which we assume a value of 10% (Huang et al., 2019). The uncertainty of using $n$-alkanes as surrogate standards for the total IVOC mass was estimated to be less than 6.0% for alkanes and 30.6% for PAHs based on the analysis of a suite of standard compounds (Table S4). The overall uncertainty for IVOCs measurement was determined to be 32.2% according to error propagation.

$$UNC_{IVOCs} = \sqrt{(\sigma^2_{sampling} + \sigma^2_{measurement})}$$

**Table S4.** List of individual n-alkanes and PAHs and their relative standard deviation (RSD, %)

| Compounds | RSD (%) | Compounds | RSD (%) |
|---|---|---|---|
| Naphthalene* | 7.8% | *n*-Dodecane* | 4.8% |
| Acenaphthene* | 3.2% | *n*-Tridecane* | 5.8% |
| Acenaphthylene* | 21.1% | *n*-Tetradecane* | 2.0% |
| 2-Methylnaphthalene* | 4.4% | *n*-Pentadecane* | 3.2% |
| 1-Methylnaphthalene* | 2.0% | *n*-Hexadecane* | 3.6% |
| 1,4-Dimethylnaphthalene* | 5.8% | *n*-Heptadecane* | 5.4% |
| Phenanthrene* | 6.7% | *n*-Octadecane* | 3.0% |
| Anthracene* | 4.9% | *n*-Nonadecane* | 2.0% |
| Fluorene* | 21.3% | *n*-Eicosane* | 2.6% |
| Fluoranthene* | 30.6% | *n*-Heneicosane* | 5.3% |
| Pristane* | 5.4% | *n*-Docosane* | 6.0% |
| Phytane* | 7.1% | | |

The manuscript has been revised as follows (Line 170-175):

"The uncertainty of the IVOCs could be ascribed to both sampling and analysis. The sampling uncertainty was assumed as 10% (Huang et al., 2019). The uncertainty of using *n*-alkanes as surrogate standards for the total IVOC mass was estimated to be less than 6.0% for alkanes and 30.6% for PAHs based on the analysis of a suite of standard compounds (SI). Therefore, combined the above uncertainty, we consider a maximum IVOCs mass uncertainty of 32.2% (SI)."

**S5 Uncertainty of IVOCs measurement** (SI line 76-86)

The uncertainty of the IVOCs could be ascribed to both sampling and analysis. When sampling, the positive/negative adsorption/desorption of

the target compounds on quartz filters/Tenax tubes (May et al., 2013) and slight flow fluctuation will cause sampling uncertainty which we assume a value of 10%(Huang et al., 2019). The uncertainty of using n-alkanes as surrogate standards for the total IVOC mass was estimated to be less than 6.0% for alkanes and 30.6% for PAHs based on the analysis of a suite of standard compounds (Table S4). The overall uncertainty for IVOCs measurement was determined to be 32.2% according to error propagation.

$$UNC_{IVOCs} = \sqrt{(\sigma^2_{sampling} + \sigma^2_{measurement})} \qquad (5)$$

**Table S4.** List of individual n-alkanes and PAHs and their relative standard deviation (RSD, %)

| Compounds | RSD (%) | Compounds | RSD (%) |
|---|---|---|---|
| Naphthalene* | 7.8% | *n*-Dodecane* | 4.8% |
| Acenaphthene* | 3.2% | *n*-Tridecane* | 5.8% |
| Acenaphthylene* | 21.1% | *n*-Tetradecane* | 2.0% |
| 2-Methylnaphthalene* | 4.4% | *n*-Pentadecane* | 3.2% |
| 1-Methylnaphthalene* | 2.0% | *n*-Hexadecane* | 3.6% |
| 1,4-Dimethylnaphthalene* | 5.8% | *n*-Heptadecane* | 5.4% |
| Phenanthrene* | 6.7% | *n*-Octadecane* | 3.0% |
| Anthracene* | 4.9% | *n*-Nonadecane* | 2.0% |
| Fluorene* | 21.3% | *n*-Eicosane* | 2.6% |
| Fluoranthene* | 30.6% | *n*-Heneicosane* | 5.3% |
| Pristane* | 5.4% | *n*-Docosane* | 6.0% |
| Phytane* | 7.1% | | |

Reference:

Huang, G., Liu, Y., Shao, M., Li, Y., Chen, Q., Zheng, Y., Wu, Z., Liu, Y., Wu, Y., Hu, M., Li, X., Lu, S., Wang, C., Liu, J., Zheng, M., and Zhu, T.: Potentially Important Contribution of Gas-Phase Oxidation of Naphthalene and Methylnaphthalene to Secondary Organic Aerosol during Haze Events in Beijing, Environmental Science & Technology, 53, 1235-1244, 10.1021/acs.est.8b04523, 2019.

May, A. A., Presto, A. A., Hennigan, C. J., Nguyen, N. T., Gordon, T. D., and Robinson, A. L.: Gas-particle partitioning of primary organic aerosol emissions: (1) Gasoline vehicle exhaust, Atmospheric Environment, 77, 128-139, https://doi.org/10.1016/j.atmosenv.2013.04.060, 2013.

*2. Line 181-183, yes, the emission factors are lower, but the gasoline consumption is higher. Isn't it the folding of both which is important for the atmospheric effect?*

Response: We thank the reviewer for the suggestions. The IVOCs emission factors we use here is a comprehensive index, which considers both the IVOCs mass and the fuel consumption. The IVOCs emission factors were calculated using carbon-mas-balance method, $EF_{IVOCs}= \frac{[\Delta IVOC]}{[\Delta CO_2]}f_c$ . Here, the $[\Delta IVOC]$ represents the background-corrected mass concentration of IVOCs, $[\Delta CO_2]$ is the background-corrected $CO_2$ concentration in the CVS expressed in units of carbon mass and fc is the measured mass fraction of carbon in the gasoline (0.82).

We agree with the reviewer that fuel consumption under different operating condition would be different. The fuel consumption at high acceleration rate (6.0 km/h/s) would be higher than that at low acceleration rate (idling). Although not emitted in IVOCs, the high consumption of the fuel would exist as other types of carbon e.g. VOCs and $CO_2$ which may also have great effects on the atmosphere. Considering the majority of the carbon emission from the exhausts is carbon dioxide (~99%), we normalize the IVOCs emission using $CO_2$ and the mass fraction of carbon in the gasoline to balance the effects of the IVOCs emission and fuel consumption so as to get a comprehensive

picture of the IVOCs emission. Therefore, in our opinion, the use of IVOCs EFs can describe the effects of acceleration rates on IVOCs emission in a modest way.

We revised manuscript so as to make it more clear to readers. The revision are as follows (line 195-205):

"Various operating conditions may cause different IVOCs emission and fuel consumption. In order to get a relative reliable comparison, what we show here is all described in IVOCs EFs which consider both IVOCs mass and the fuel consumption. Among all of the factors, acceleration rate has the largest influence on the IVOC EFs. The fuel consumption at high acceleration rate (6.0 km/h/s) would be higher than that at low acceleration rate (idling). Although not emitted in IVOCs, the high consumption of the fuel would exist as other types of carbon e.g. VOCs and $CO_2$ which may also have great effects on the atmosphere. Therefore, the usage of IVOCs EFs can moderately balance the effects of the IVOCs emission and fuel consumption and get a comprehensive comparison among different acceleration rates."

**3. Line 251 / caption Figure 3, Figure 3 needs a better introduction and captions especially introducing the Chinese E10 trace.**

Response: We thank the reviewer for the suggestion. We extend the figure caption to make it easier for the readers.

Studies performed in US used commercial US gasoline as fuel, which

contained 10% v/v ethanol, i.e. E10 fuel. Therefore, all the US fuel/US unburned fuel/ US gasoline in the main text means US E10 fuel. In addition, Zhao et al. (2016) and Lu et al. (2018) found that consistent distribution of US fuel and exhaust, so in Figure 3, the US gasoline vehicle exhaust can represent the volatility distribution of its unburned fuel distribution as well. As a result, we compare our exhaust and E10 fuel with US exhaust and E10 fuel to get a comparative study. In the revised manuscript we discussed the reason why we use Chinese E10 fuel to do the fuel comparison with US fuel, and explained why we just used curve to represent the US vehicle exhaust and the unburned fuel so as to make the readers more clear of our view. In addition, we modified the figure caption in Figure 3 as well to avoid confusion.

The manuscript has been revised as follows (line 373-375):

"As the tests of US vehicles were all performed using California commercial fuel, which is, in fact, E10 fuel. Therefore, in this study, the US (unburned) fuel or US gasoline means E10. Lu et al. …distribution. As a result, in Figure 3, we use US gasoline vehicle exhaust to both represent the exhaust and the unburned (E10) fuel and compare the Chinese E10 fuel with US fuel to get a comparative study."

The figure caption has been modified as follows (line 843-854):

**Figure 3**. Comparison of IVOC volatility distributions of Chinses gasoline vehicle exhaust, US gasoline vehicle exhaust, and Chinses E10

fuel. The box-plot represents the Chinses gasoline vehicle exhaust. The boxes represent the 25th and 75th percentiles with the centerline being the median. The whiskers are the 10th and 90th percentiles. Red solid circles represent IVOC fractions of US vehicle exhaust (Zhao et al., 2016). Blue hollow triangles represent the IVOCs volatility distribution of Chinese E10 fuel. As all the studies performed in US used commercial US gasoline as fuel, which contained 10% v/v ethanol, i.e. E10 fuel. Therefore, we compare the Chinese E10 with US fuel to get a consistent comparison. Also, we should note that Zhao et al. (2016) and Lu et al. (2018) found that consistent distribution of US fuel and exhaust, so in this figure, the US gasoline vehicle exhaust can represent the volatility distribution of its unburned fuel distribution as well.

References:

Lu, Q., Zhao, Y., and Robinson, A. L.: Comprehensive organic emission profiles for gasoline, diesel, and gas-turbine engines including intermediate and semi-volatile organic compound emissions, Atmospheric Chemistry and Physics, 18, 17637-17654, 2018.
Zhao, Y., Nguyen, N. T., Presto, A. A., Hennigan, C. J., May, A. A., and Robinson, A. L.: Intermediate Volatility Organic Compound Emissions from On-Road Gasoline Vehicles and Small Off-Road Gasoline Engines, Environmental Science & Technology, 50, 4554-4563, 10.1021/acs.est.5b06247, 2016.

*4. Line 255-265, I suggest always (4x) to refer to the panels in FigureS5 in order to relate the statements to the plots. It is easier for the reader.*

Response: Thank you for your comment. We have double checked the manuscript to relate the statements to the plots in order to make it

more clear to the readers. The manuscript has been altered accordingly.

**5. Line 296-298, these are results from only one vehicle, therefore I suggest to formulate the conclusions a bit more careful.**

Response: Thank you for your comment. We agree with the reviewer that singular vehicle might be not enough to support the conclusion we formulate in the manuscript. We have added analysis and references to support our ideas and the relevant sentences has been altered more carefully to get a relative moderate conclusion. The details are as follows (line 323-343):

In addition, we compared our results with that from European vehicles, and found that the $NO_x$ and THC EFs for the tested vehicle were lower than Euro 5 gasoline vehicle, while the PM EF was higher (Fontaras et al., 2014). This suggests that compared with US and European vehicles, the stringent emission implemented by Chinese government have been effective at controlling $NO_x$ and THC, but might be inefficient to PM emissions. For past 30 years, Chinese government has adopted a series of emission control policies and measures for light-duty vehicles, including implementation of emission standards for new vehicles promotion of sustainable transportation and alternative fuel vehicles, and traffic management programs (Wu et al., 2017; Zhang et al., 2014). Wu et al. (2017) summarizes the implementation of the vehicle

control policies in China, which shows the control for the vehicular pollutants is becoming stricter step by step. For example, the $NO_x$ emission standard changed from 0.15 g $km^{-1}$ to 0.035 g $km^{-1}$ while the standard changed from China III to China VI. Different from $NO_x$ and THC which has been controlled since China III, only when in 2017, China V standard first introduced the control of PM into the emission control scope. Yang et al. (2020)investigated the effects of gasoline upgrade policy on migrating the PM pollution in China and found that there's no much space for significantly reducing the PM concentration by simply improving the gasoline quality. Therefore, for PM control, more policies i.e. developing cleaner alternatives to fossil fuels, replacing traditional vehicles with new-energy and building developed public transport system should be done.

References:

Fontaras, G., Franco, V., Dilara, P., Martini, G., and Manfredi, U.: Development and review of Euro 5 passenger car emission factors based on experimental results over various driving cycles, Science of The Total Environment, 468-469, 1034-1042, https://doi.org/10.1016/j.scitotenv.2013.09.043, 2014.

Wu, Y., Zhang, S., Hao, J., Liu, H., Wu, X., Hu, J., Walsh, M. P., Wallington, T. J., Zhang, K. M., and Stevanovic, S.: On-road vehicle emissions and their control in China: A review and outlook, Science of The Total Environment, 574, 332-349, 2017.

Yang, G., Zhang, Y., and Li, X.: Impact of gasoline upgrade policy on particulate matter pollution in China, Journal of Cleaner Production, 262, 121336, 10.1016/j.jclepro.2020.121336, 2020.

Zhang, S., Wu, Y., Wu, X., Li, M., Ge, Y., Liang, B., Xu, Y., Zhou, Y., Liu, H., Fu, L., and Hao, J.: Historic and future trends of vehicle emissions in Beijing, 1998–2020: A policy assessment for the most stringent vehicle emission control program in China, Atmospheric Environment, 89, 216-229, https://doi.org/10.1016/j.atmosenv.2013.12.002, 2014.

**6. Line 337-340, I cannot see this in Figure 3. Or should I compare to E10 fuel? However, why E10 fuel then? As mentioned already, Figure 3 needs better explanations.**

Response: Thank you for your comment.

Sorry for the ambiguous description. We have modified the manuscript both in the figure caption and the contents in the corresponding part (line 373-386).

**7. Line 410-412, I am sorry, but this sentence does not make sense to me. (grammar?) Please, rephrase.**

Response: Thank you for your comment. We want to emphasize the great SOA formation potential of the E10 fuel using enhancement factor (SOA-to-POA ratio) as a metric. We feel sorry for the ambiguous description of the sentence. We have modified the manuscript so as to make the sentence more clear to readers. The manuscript has been altered as follows (line 457-463):

"Though the POA emission for gasoline-fueled vehicle was higher than that fueled by E10, comparable SOA formation is estimated using gasoline and E10 as fuel. That means, the OA enhancement factor for E10 is higher than that of gasoline. This suggests that although the ongoing policy of ethanol gasoline will not exacerbate the POA emission in China, the SOA formation of E10 could not be neglected due to its

high SOA enhancement capacity. Therefore, more research should be done to evaluate the effectiveness of using E10 as surrogate to reduce the air pollution in China."

*8. Line 421/Figure S10, wouldn't it be good to indicate the contribution of the classes to the emission (Figure S4). Or bring Figure S4 and S10 closer together. I guess the aromatics in Figure S4 contain also the single ring aromatics. That would mean herethat aromatics are over-effective in SOA formation.*

Response: Thank you for your comment. There might be some misunderstanding due to our omission in figure caption/contents.

In fact the aromatics in Figure S4 represent the polycyclic aromatic hydrocarbons (PAHs) we have in the IVOCs standards, i.e. naphthalene, acenaphthene, acenaphthylene, 2-methyl naphthalene, 1-methylnaphthalene, 1,4-dimethylnaphthalene, phenanthrene, anthracene, fluorene, fluoranthene and pyrene (Table S1) while the single-ring aromatics in Figure S10 represent the single-ring aromatics in VOCs range, i.e. benzene, toluene, o/m/p-xylene, ethylbenzene, styrene, isopropyl-benzene, n-propyl-benzene, o/m/p-ethyl-toluene, 1,3,5-trimethyl-benzene, 1,2,4-trimethyl-benzene and 1,2,3-trimethyl-benzene. Therefore, according to our classification, the single-ring-aromatics falling into IVOCs range are divided into the

unspeciated cyclic compounds.

Figure S4 and S10 were described in different parts of our manuscript to clarify our view. Figure S4 aims to present the IVOCs emissions from the exhaust which belong to the primary emission from pipeline exhaust while Figure S10 aims to estimate the contribution of different compounds (IVOCs+VOCs) to the SOA formation which belong to the secondary formation part. Therefore, we think the arrangement now might be more suitable.

We have modified the manuscript both in figure caption and in the main contents to avoid misunderstanding.

***9. Line 434-436: ". . .and then keeps constant after~24 h." No, I would say it does not become constant within the first 48h. I could agree with a formulation "levels off after 30h", or "the curves flatten after 24.-30h".***

Response: Thank you for your comment. We agree with the reviewer that "keeps constant after ~24h" may not be suitable, we have changed the description according to the reviewer's suggestion. The original sentence "In general, SOA exceeds POA after first a few hours of oxidation, and then keeps constant after ~24 h" was replaced by "In general, SOA exceeds POA after first a few hours of oxidation, and then levels off after 30 h." (line 484)

**10.** ***Line 444 and Table 1, could you show the quality of your fits? E.g. plotted into Figure S11?***

Response: Thank you for your comment.

We have plotted the curves in Figure S13 to show the quality of our fits. Figure S13 (a)-(d) represent the fits of SOA/POA versus photochemical age under different $NO_x$ condition: (a) low $NO_x$; (b) high $NO_x$ at an OA loading of 10 $\mu g \cdot m^{-3}$; (c) high $NO_x$ at an OA loading of 20 $\mu g \cdot m^{-3}$; (d) high $NO_x$ at an OA loading of 80 $\mu g \cdot m^{-3}$. Figure S13 (e)-(h) show the fits of coefficients, taken single-ring aromatics, unspeciated $b$-alkanes, unspeciated cyclic compounds and $n$-alkanes as examples. We could see from the figure that our fits could moderately reflect our simulation results.

The manuscript and the supporting information has been revised accordingly.

[Figure]

**Figure S13.** Fits of SOA/POA versus photochemical age at different NOx condition (a)-(d): (a)

low NO$_x$ condition; (b) at an OA loading of 10 μg·m$^{-3}$ under high NO$_x$ condition; (c) at an OA

loading of 20 μg·m$^{-3}$ under high NO$_x$ condition; (d) at an OA loading of 80 μg·m$^{-3}$ under high NO$_x$ condition. Fits of coefficients (e)-(h): (e) single ring aromatics; (f) unspeciated cyclic compounds; (g) unspeciated *b*-alkanes and (h) *n*-alkanes.

**11. Line 758 / Figure 5, explain the "balls" in the caption**

Response: Thank you for your comment. We have added the description of balls in the caption which can be could as follows: "The blue circles represent the SOA-to-POA ratio after 48 h of photooxidation (right axis)." (line 868-869)

**Errors:**

**1. Line 27, B14-B16 compounds, this notation cannot be used here as it is not explained.**

Response: Thank you for your comment. We have modified the abstract according to the reviewer's comment. We have added the explanation of the B$_{14}$-B$_{16}$ (retention time bins corresponding to C$_{14}$-C$_{16}$ *n*-alkanes) to the abstract. (line 28-29)

**2. Line 30, I suggest to use "did" have instead of "could" have**

Response: Thank you for your comment. We have changed the word in the manuscript from "could" to "did". (line 31)

**3. Lines 35, I would replace "vehicle" by "the tested vehicle", or so. In**

*any case "the" is missing.*

Response: Thank you for your comment. We have made correction according to the Reviewer's comments. "The tested vehicle" is added into Line 38.

*4. Line 58 and many more instances: a blank is missing in the reference listings.*

Response: Thank you for your comment. We have modified the endnote output style to solve this problem. The manuscript has been altered accordingly.

*5. Line 104, it was only one vehicle, so please use singular*

Response: Thank you for your comment. We have modified the manuscript according to the reviewer's comment. The original sentence "Prior to tests, vehicles were preconditioned with an overnight soak, without evaporative canister purge." was replaced by "Prior to tests, the tested vehicle was preconditioned with an overnight soak, without evaporative canister purge."

*6. Line 120/121, either articles or use of plural for "quartz filter(s)" and "TENAX tube(s)"*

Response: Thank you for your comment. The manuscript has been

altered accordingly.

**7. Line 129, "a" gas chromatograph mass spectrometer or mass spectrometr"y"**

Response: Thank you for your comment. We have modified the sentence according to the reviewer's comment by adding a and changing spectrometer to spectrometry. The original sentence "Quartz filters and Tenax tubes were analyzed using gas chromatography/mass spectrometer (Agilent 6890GC/5975MS)" was changed to "Quartz filters and Tenax tubes were analyzed using a gas chromatography/mass spectrometry (Agilent 6890GC/5975MS)". (line 140-141)

**8. Line135, you can skip "in the literature"**

Response: Thanks for the comment. We have deleted "in the literature" from that sentence.

**9. Line 144, please explain the notation SUUMA**

Response: Thanks for the comment. We have added the description of SUMMA canister into the manuscript. The original sentence "VOCs were sampled in SUUMA canisters and analyzed using GC-MS with a flame ionization detector." was changed to "VOCs were sampled in

SUUMA® polished stainless steel canisters and analyzed using GC-MS with a flame ionization detector." (line 153-154)

**10.   Line 209, . . . found "that the" NECD cycle. . ., or so**

Response: Thank you for your comment. We have modified the sentence according to the reviewer's suggestion. (line 231)

**11.   Line 251 . . ."show". . ."over" the 11 retention time bins.**

Response: Thank you for your comment. We have adjusted the sentence according to the reviewer's suggestion. (line 271)

**12.   Line 296, . . .emission "measures" implemented. . ., or so**

Response: Thank you for your comment. We have modified the manuscript accordingly.

**13.   Line 300, Chinese regulations "may" also appear. . .**

Response: Thank you for your comment. We have modified the sentence according to the reviewer's suggestion.

**14.   Line 305, Figure S8, I guess**

Response: Thank you for your comment. We feel very sorry for our carelessness, the wrong figure number has been altered in the

manuscript.

**15. Line 322, ..has a similar IVOC volatility distribution "as" the unburned gasoline. . .**

Response: We thank the reviewer for the comment. We have corrected the sentence according to the reviewer's suggestion. The sentence "US vehicle exhaust has a similar IVOC volatility distribution to the unburned gasoline, indicating…" has been changed to "US vehicle exhaust has a similar IVOC volatility distribution as the unburned gasoline, indicating…".

**16. Line 366 and more place, typo in the word "Chinese", please double check and correct.**

Response: Thank you for your comment. We feel very sorry for the careless writing and have corrected the spelling in the manuscript.

**17. Line 415, I would start a new paragraph here, beginning with "Cold start. . ."**

Response: Thank you for your comment. We agree with the reviewer that a new paragraph beginning with "Cold start…" would be better. The manuscript has been altered according to the reviewer's opinion.

**18. Line 460, Compared with US LEV-2 gasoline vehicle"s", "the" China V vehicle emits three times "more" IVOCs. Three suggested changes in "" "".**

Response: Thank you for your comment. We have modified the manuscript according to the reviewer's comment. The original sentence "Compared with US LEV-2 gasoline vehicle, China V vehicle emits three times higher IVOCs." was altered to "Compared with US LEV-2 gasoline vehicles, the China V vehicle emits three times more IVOCs."

**19. Line 463-464, . . .IVOCs could act "as" more important SOA precursors. . .**

**20. I found typos in the supplement (which has no line numbering and page numbers),which you can find by searching: hot-start; Zhao et al. (ref.) => Zhao et al. (2016); "b-alkane" is double.**

Response: Thank you for your comment. We feel so sorry for the careless writing, we have modified the relevant content in the supplement.

---

## Author Comment (AC2) · 9 Jan 2021

We thank the referee for the careful reviews and suggestions. Following is our response to the comments:

➤ *Referee #2:*

*General comments: The manuscript presents novel data regarding IVOC emission factors for a gasoline/E10 Chinese vehicle, that meets China V standard. Methods are sound, the language is cogent and very easy to follow. The presentation of the results is very clear and the main findings are thoroughly discussed and compared to previous literature, considering differences and similarities with US-based data. As the paper entails important implications for both the scientific community and policymakers, I recommend final publication after minor revisions. The following comments are mostly aimed to improve the readability, interpretability and usefulness of the study for future work.*

*Specific comments*

*1. To facilitate the use of your new data in modeling studies using the Volatility Basis Set (VBS) scheme, I would recommend to present the volatility distribution data also in terms of saturation concentration bins, in a similar way to Zhao et al. [2016] (Figure 4). Also, it would be convenient if you can report a Table, maybe in the SI, reporting the mass fraction distribution of organics for each saturation concentration bin (e.g. Table S5 in Zhao et al. [2016]). These values are usually a key input for the VBS schemes in state-of-the-art numerical models. In*

*addition to this, I would suggest to report the median IVOC-to-THC ratio in the abstract as well, as that is key information for modelers.*

Response: Thank you for your comment. We have presented the volatility distribution data both in figure and table in the manuscript. The revision was as follows: "Considering the similarity of volatility distribution for different conditions and the importance of the volatility distribution in model input for SOA simulation, Figure S6 and Table S3 present the volatility distribution of SVOC and IVOC emissions from the tested China V gasoline vehicle, using effective saturation concentration (C*) as classification: IVOCs (C*=300-3$\times10^6$ $\mu$g·m$^{-3}$), SVOCs (C*=0.3-300 $\mu$g·m$^{-3}$ ). IVOCs are the dominant part of the low volatility organics (IVOCs+SVOCs), with a median contribution of ~95%."

[Figure]

**Figure S6.** Volatility distribution of organics measured by GC/MS of adsorbent tubes

collected during all the tests for the tested China V gasoline vehicle. The boxes represent the 75$^{th}$ and 25$^{th}$ percentiles, with the centerline being the median. The whiskers are the highest and lowest values.

**Table S3.** Median volatility distribution of IVOCs, SVOCs obtained by GC-MS analysis of Tenax tubes, C* 100 to 106 μg m$^{-3}$) as a function of effective saturation concentration (C*, μg·m$^{-3}$) at 298 K.

| Log (C$^*$) | 50$^{th}$ |
|---|---|
| 0 | 0.009 |
| 1 | 0.019 |
| 2 | 0.018 |
| 3 | 0.027 |
| 4 | 0.095 |
| 5 | 0.206 |
| 6 | 0.624 |

*2. In the "atmospheric implications" section, I would suggest to at least mention the possible limitations of the study, and maybe possible future directions. One example could be the fact that only one vehicle was tested (China V), and different values might be obtained for different vehicles (even vehicles that meet the China V emission standard), implying that the total uncertainty associated with the estimated emission factors might be larger. Also, when discussing why your estimate of total IVOC emissions in China is conservative (lines 476-480), can you report what is the current percentage of vehicles that meet the China V standard in the Chinese car fleet? This would help the reader understanding the extent of the implications of the assumption made in estimating that the total IVOC emissions in China*

*are 30 Gg.*

Response: Thank you for your comment. The representativeness of singular tested car will cause uncertainties which restricts the future model simulation. Therefore, we mentioned repeatedly in the manuscript that our result is only based on the tested China V car. We also compared our results with US vehicles in different controlling stages to verify the representativeness of our tested vehicle. Though some uncertainty may exist, the tested car still has its representativeness. More importantly, the aim of this study is to compare the IVOCs emissions under different conditions so as to provide effective suggestions for developing new technologies to reduce pollution from vehicles and making controlling policies in future vehicular management. For this reason, we think singular tested vehicle can consistently evaluate the effects of different factors on IVOC emissions.

We agree with the reviewer that possible limitations of the studies and the current situation of Chinese gasoline vehicles should be mentioned in the implication part.

The Atmospheric Implications part has been altered accordingly. More details could be found as follows (line 542-550):

"Though we have discussed the influences of different operating conditions on IVOC emissions and SOA formation for the tested China V gasoline vehicle, due to the singular vehicle tests of our study, more

research i.e. vehicles meeting different emission standards, different engines should be performed both to testify the accuracy of our research and to get a full understanding of the IVOC emission inventory for Chinese gasoline vehicles. Furthermore, advanced measurement techniques e.g. GC×GC-MS and chemical ionization mass spectrometry (CIMS) should be used to obtain a comprehensive molecular-level picture of the total organics so as to reduce the uncertainties associated with the measurements and models."

We have also included the current percentage of vehicles that meet the China V standard in the Chinese car fleet into the manuscript according to the reviewer's comment. Details could be found in the revised manuscript (line 518-528).

"Till the end of 2018, the total vehicle population in China reached 0.327 billion, with automobiles contributing 61% (0.24 billion). Of all the automobiles, gasoline-fueled car took the dominant (88.1%)…. According to the statistics from the Ministry of Ecology and Environment, only 30.9% of the vehicles in 2018 meet the standards of China V. Indeed, higher percentage of pre-China V e.g. China I-IV standard cars will cause more IVOCs emission. In addition, the IVOC/NMHC ratio of diesel vehicles could be much higher than that of the gasoline vehicles (Zhao et al., 2016, 2015). This may also lead to an underestimation."

***3. In Section 3.3, you mention several times that recent Chinese regulations failed in controlling PM emissions (and IVOC emissions as well), whereas they were effective for NOx and THC, according to your data. Can you expand on that? Which regulations did they implement? Why do you think they were ineffective for PM and IVOCs but effective for NOx and THC? Maybe some additional references might help – Expanding the discussion on this point can be useful to guide policymaking.***

Response: Thank your comment.

The reason why we get this conclusion is that roughly the standard of China V is comparable to US LEV-2 vehicles. While $NO_x$ and THC EFs fall into the range of US LEV-2 vehicle, PM and IVOC EFs lie in the range of pre-LEV and LEV-1. Therefore, in our opinion, compared with $NO_x$ and THC, the effectiveness of the PM and IVOCs control is (at least) not as good as that for $NO_x$ and THC.

We agree with the reviewer that expansion of the discussion will be more useful both for readers and for policy making. The manuscript has been altered as following (line 323-343):

"In addition, we compared our results with that from European vehicles, and found that the $NO_x$ and THC EFs for the tested vehicle were lower than Euro 5 gasoline vehicle, while the PM EF was higher (Fontaras et al., 2014). This suggests that compared with US and

European vehicles, the stringent emission implemented by Chinese government have been effective at controlling NOx and THC, but might be inefficient to PM emissions. For past 30 years, Chinese government has adopted a series of emission control policies and measures for light-duty vehicles, including implementation of emission standards for new vehicles promotion of sustainable transportation and alternative fuel vehicles, and traffic management programs (Wu et al., 2017; Zhang et al., 2014). Wu et al. (2017) summarizes the implementation of the vehicle control policies in China, which shows the control for the vehicular pollutants is becoming stricter step by step. For example, the $NO_x$ emission standard changed from 0.15 g $km^{-1}$ to 0.035 g $km^{-1}$ while the standard changed from China III to China VI. Different from NOx and THC which has been controlled since China III, only when in 2017, China V standard first introduced the control of PM into the emission control scope. Yang et al. (2020)investigated the effects of gasoline upgrade policy on migrating the PM pollution in China and found that there's no much space for significantly reducing the PM concentration by simply improving the gasoline quality. Therefore, for PM control, more policies i.e. developing cleaner alternatives to fossil fuels, replacing traditional vehicles with new-energy and building developed public transport system should be done."

References:

Fontaras, G., Franco, V., Dilara, P., Martini, G., and Manfredi, U.: Development and review of Euro 5 passenger car emission factors based on experimental results over various driving cycles, Science of The Total Environment, 468-469, 1034-1042, https://doi.org/10.1016/j.scitotenv.2013.09.043, 2014.

Wu, Y., Zhang, S., Hao, J., Liu, H., Wu, X., Hu, J., Walsh, M. P., Wallington, T. J., Zhang, K. M., and Stevanovic, S.: On-road vehicle emissions and their control in China: A review and outlook, Science of The Total Environment, 574, 332-349, 2017.

Yang, G., Zhang, Y., and Li, X.: Impact of gasoline upgrade policy on particulate matter pollution in China, Journal of Cleaner Production, 262, 121336, 10.1016/j.jclepro.2020.121336, 2020.

Zhang, S., Wu, Y., Wu, X., Li, M., Ge, Y., Liang, B., Xu, Y., Zhou, Y., Liu, H., Fu, L., and Hao, J.: Historic and future trends of vehicle emissions in Beijing, 1998–2020: A policy assessment for the most stringent vehicle emission control program in China, Atmospheric Environment, 89, 216-229, https://doi.org/10.1016/j.atmosenv.2013.12.002, 2014.

*4. Some claims in the introduction can be better substantiated by referencing previous literature. E.g. lines 58-59 "A large discrepancy remains between modeled and measured SOA. One possible reason is missing SOA precursors." Two recent modeling works that discussed these two points are Giani et al. [2019] in Europe and [Huang et al., 2020] in China, and I suggest to add a citation to strengthen your claims. In the introduction, I would also stress the point that understanding and characterizing IVOC emissions, as well as their volatility distributions, is crucial for improving numerical models that aim to predict OA.*

*References*

*Giani, P., A. Balzarini, G. Pirovano, S. Gilardoni, M. Paglione, C.*

*Colombi, V. L. Gianelle, C. A. Belis, V. Poluzzi, and G. Lonati (2019), Influence of semi-and intermediatevolatile organic compounds (S/IVOC) parameterizations, volatility distributions and aging schemes on organic aerosol modelling in winter conditions, Atmospheric environment, 213, 11-24.*

*Huang, L., Q. Wang, Y. Wang, C. Emery, A. Zhu, Y. Zhu, S. Yin, G. Yarwood, K. Zhang, and L. Li (2020), Simulation of secondary organic aerosol over the Yangtze River Delta region: The impacts from the emissions of intermediate volatility organic compounds and the SOA modeling framework, Atmospheric Environment, 118079.*

*Zhao, Y., N. T. Nguyen, A. A. Presto, C. J. Hennigan, A. A. May, and A. L. Robinson (2016), Intermediate volatility organic compound emissions from on-road gasoline vehicles and small off-road gasoline engines, Environmental science & technology, 50(8), 4554-4563.*

Response: Thank you for your comment. We have modified the manuscript according to the reviewer's comment. Details can be found as following (line 71-82):

"Recent model studies have shown that adding IVOC emissions into different models will greatly improve the SOA simulation results. For example, Giani et al. (2019) found a considerable OA enhancement in Po Valley (Northern Italy) when applying new S/IVOCs emission estimates

and the new volatility distributions into CAMx, in which the improvement in SOA mainly due to the revised IVOC emissions. Huang et al. (2020) found a similar enhancement in SOA simulation for Yangtze River Delta (Southeast China) region when adding IVOC emissions into CAMx. They also show the importance of volatility distribution and emission parameterization for the model simulation. Therefore, understanding and characterizing IVOC emissions, as well as their volatility distributions, is crucial for improving numerical models that aim to predict OA."

Reference:

Giani, P., Balzarini, A., Pirovano, G., Gilardoni, S., Paglione, M., Colombi, C., Gianelle, V. L., Belis, C. A., Poluzzi, V., and Lonati, G.: Influence of semi- and intermediate-volatile organic compounds (S/IVOC) parameterizations, volatility distributions and aging schemes on organic aerosol modelling in winter conditions, Atmospheric Environment, 213, 11-24, 10.1016/j.atmosenv.2019.05.061, 2019.
Huang, G., Liu, Y., Shao, M., Li, Y., Chen, Q., Zheng, Y., Wu, Z., Liu, Y., Wu, Y., Hu, M., Li, X., Lu, S., Wang, C., Liu, J., Zheng, M., and Zhu, T.: Potentially Important Contribution of Gas-Phase Oxidation of Naphthalene and Methylnaphthalene to Secondary Organic Aerosol during Haze Events in Beijing, Environmental Science & Technology, 53, 1235-1244, 10.1021/acs.est.8b04523, 2019.
Huang, L., Wang, Q., Wang, Y., Emery, C., Zhu, A., Zhu, Y., Yin, S., Yarwood, G., Zhang, K., and Li, L.: Simulation of secondary organic aerosol over the Yangtze River Delta region: The impacts from the emissions of intermediate volatility organic compounds and the SOA modeling framework, Atmospheric Environment, 118079, 10.1016/j.atmosenv.2020.118079, 2020.

**5. I am a little skeptical about the parametrization presented in Section 3.5, which seems somewhat arbitrary. Does the logarithmic curve have some sort of physical insight or is it based only on the shape of the calculated curve? Why not using something like k-exp(. . .) as in the**

*actual model used to derive that curve (Equation in Section 3.4), also because you're claiming that after 24h SOA/POA is approximately constant? The other concern that I have is that there are a lot of parameters to be estimated (9 in high-NOx conditions), which might cause overfitting to your data, thus losing generalizability. Is it a specific reason why you're using so many parameters? Is there a way of having a simpler parametrizations with similar fit performance? If so, a simpler model (i.e. with less parameters) should be preferred. I would suggest that at least you should better justify your choices for the proposed parametrization in Section 3.5. I believe that Section 3.5 can be largely improved, either by better substantiating your choices or performing some further calculations (that might exceed the scope of the paper, though).*

Response: We thank the reviewer for the comment.

We agree with the reviewer that the introduction of more parameters will bring more uncertainties. The logarithmic curve was chosen due to the shape and the fitting results. There might not be accurate physical meaning of these parameters or we should study it deeper in future studies. Actually, in this part, we just try to establish some empirical formulas to help understanding the SOA formation of the gaseous precursors, especially the missing IVOCs. As previous studies have shown that the SOA yields for different IVOCs and VOCs depend

strongly on the OA loading under high $NO_x$ conditions, which can have significant influences on the model simulation results. In order to get a comprehensive understanding of these effects, we introduce these parameters. According to these seemingly complicated parameters, we could roughly estimate the SOA formation based on the OA concentration and the photochemical age which might, at least, provide a way for us to calculate the SOA formation.

More research should be done to testify the effectiveness and representativeness of these empirical equations. In fact, we are now expanding our research to do more sensitive analysis of the SOA formation using CMAQ coupled with VBS which might answer the reviewer's comment. This work is now in progress and might be submitted to ACP in several months.

**6. What are the dots in Figure 5? Please explain in the caption. (I'm assuming is the SOA/POA ratio to be read on the right scale?)**

Response: Thank you for your comment. We have added the description of the blue dots in figure caption which can be found as follows: "The blue circles represent the SOA-to-POA ratio after 48 h of photooxidation (right axis)".